# Acidification suppresses the natural capacity of soil microbiome to fight pathogenic *Fusarium* infections

Xiaogang Li [1,2,11], Dele Chen[3,4,11], Víctor J. Carrión[5,6,7,8], Daniel Revillini [9], Shan Yin[4], Yuanhua Dong[3], Taolin Zhang[3], Xingxiang Wang[3,10] & Manuel Delgado-Baquerizo [9]

Soil-borne pathogens pose a major threat to food production worldwide, particularly under global change and with growing populations. Yet, we still know very little about how the soil microbiome regulates the abundance of soil pathogens and their impact on plant health. Here we combined field surveys with experiments to investigate the relationships of soil properties and the structure and function of the soil microbiome with contrasting plant health outcomes. We find that soil acidification largely impacts bacterial communities and reduces the capacity of soils to combat fungal pathogens. In vitro assays with microbiomes from acidified soils further highlight a declined ability to suppress *Fusarium*, a globally important plant pathogen. Similarly, when we inoculate healthy plants with an acidified soil microbiome, we show a greatly reduced capacity to prevent pathogen invasion. Finally, metagenome sequencing of the soil microbiome and untargeted metabolomics reveals a down regulation of genes associated with the synthesis of sulfur compounds and reduction of key traits related to sulfur metabolism in acidic soils. Our findings suggest that changes in the soil microbiome and disruption of specific microbial processes induced by soil acidification can play a critical role for plant health.

Soil microbiomes play crucial roles in regulating plant growth and health[1]. Soil-borne pathogens that induce host disease have become a critical factor negatively affecting yields in agriculture globally. These pathogens are projected to increase under climate change and thus represent a major challenge to promote food production for a continuously growing global human population[2,3]. Globally distributed soil-borne pathogens, such as *Fusarium* can gravely impact the roots of plants[4]. The soil microbiome has been shown to reduce plant pathogen efficacy and represents a biological barrier against infection[5]. Indigenous microbes can suppress the growth of aggressive

[1]State Key Laboratory of Tree Genetics and Breeding, College of Ecology and Environment, Nanjing Forestry University, Nanjing, China. [2]Co-Innovation Center for Sustainable Forestry in Southern China, Nanjing Forestry University, Nanjing, China. [3]Key Laboratory of Soil Environment and Pollution Remediation, Institute of Soil Science, Chinese Academy of Sciences, Nanjing, China. [4]School of Agriculture and Biology, Shanghai Jiao Tong University, Shanghai Yangtze River Delta Eco-Environmental Change and Management Observation and Research Station, Ministry of Science and Technology, Shanghai, China. [5]Microbial Biotechnology, Institute of Biology, Leiden University, Leiden, the Netherlands. [6]Departamento de Microbiología, Facultad de Ciencias, Campus Universitario de Teatinos s/n, Universidad de Málaga, Málaga, Spain. [7]Department of Microbial Ecology, Netherlands Institute of Ecology (NIOO-KNAW), Wageningen, The Netherlands. [8]Instituto de Hortofruticultura Subtropical y Mediterránea La Mayora (IHSM) UMA-CSIC, 29010 Málaga, Spain. [9]Laboratorio de Biodiversidad y Funcionamiento Ecosistémico, Instituto de Recursos Naturales y Agrobiología de Sevilla (IRNAS), CSIC, Sevilla, Spain. [10]Ecological Experimental Station of Red Soil, Chinese Academy of Sciences, Yingtan, China. [11]These authors contributed equally: Xiaogang Li, Dele Chen. ✉e-mail: xxwang@issas.ac.cn; M.delgado.baquerizo@csic.es

pathogens through mechanisms, such as nutrient competition or substance antagonism[6,7]. Disease-suppressive soils can also lead to reduced pathogen susceptibility for the next generation of plants, known as the 'soil-borne legacy'[8,9]. Soil legacies help plants tolerate changing climates, extending beyond their influence on pathogens alone[10]. Despite all this, the environmental factors regulating the capacity of healthy soils to suppress pathogen development is virtually unknown. Edaphic properties largely impacting the soil microbiome, such as acidification could help explain the inherent disease-suppressive capacities of healthy soils[11]. However, experiments supporting this hypothesis are lacking.

Long-term evolution and cooperation among taxa in soil have produced robust natural microbial communities able to endure a wide and complex set of external pressures, including pathogen invasion[12,13]. Various microbial biocontrol agents isolated from soil can prevent pathogenesis through parasitism, predation or interference, and they have developed intricate biochemical signaling to reduce the pathogenicity of soil-borne pathogens[14,15]. Unfortunately, low colonization rates and limited functional expression of beneficial microorganism inoculations in soil have led to indeterminate microbiome-mediated pathogen prevention in the field[16]. For this reason, identifying complex microbial community interactions may be the best way to understand their capacity to inhibit pathogens[12,17]. The complementary effect of diverse communities in resource utilization and the synergistic effect of antagonistic substances reveal a community advantage for the suppression of soil-borne pathogens[6,7,18]. However, few studies have been carried out to evaluate the effect of natural microbial communities on the growth, abundance or efficacy of soil-borne pathogens. Disturbance of the soil environment can directly affect microbial community structure and functioning via regulating niche differentiation[19–21]. We posit that soil degradation impacts on the soil microbiome can have critical consequences for plant health by 'opening the door' to important soil-borne pathogen infections[22,23]. To fully address this hypothesis, a complex set of experimental systems that directly observe and test the role of microbiome mediation in plant health-pathogen relationships was required.

Here, we combined field surveys with multiple experiments to investigate the structure and function of the soil microbiome associated with contrasting crop health using peanuts (*Arachis hypogaea*) as a model plant (Fig. 1). We explicitly investigated what environmental conditions in soil support disease-suppressive capacities. Moreover, we employed shotgun metagenomic sequencing and untargeted metabolomics across a range of soil acidification levels to explore microbial functional capacity to reduce pathogenicity and promote plant health by analyzing the enrichment of important microbial functional pathways and identifying specific soil microbiome metabolites. Our work has overarching implications for land management, sustainable agriculture, and food production for growing populations as we progress through the Anthropocene.

## Results

### Soil acidification aggravates plant health

Our study included a wide variation in key soil properties such as soil textures, soil nutrients, soil organic matter and pH: 19–46% clay and 6–52% sand, 0.48–1.26% organic carbon content, and pH 4.1 to 6.8. Other soil properties varied as well (Supplementary Fig. 4, Supplementary Fig. 5). To analyze the correlation between soil properties and plant health, Spearman correlations were performed. We observed that among all tested properties, soil pH was positively correlated with plant height, while ammonium content revealed an opposite trend ($P < 0.001$). Notably, pH was the sole factor significantly correlated to plant height ($R = 0.33$, $P < 0.001$, $R^2 = 0.11$), root length ($R = 0.45$, $P < 0.001$, $R^2 = 0.20$) and root rot ($R = -0.51$, $P < 0.001$, $R^2 = 0.26$) (Fig. 2a). We then conducted a stepwise regression model to further quantify the contribution of multiple environmental factors to peanut

root rot (Fig. 2b). In this analysis, we selected 16 soil properties as explanatory variables, but only five of these soil properties were selected in the final model, explaining 36.1% of the variation in plant disease (F-statistic: 21.24 on 5 and 174 degrees of freedom, $P < 0.001$). Consistent with previous analyses, pH was the most significant variable explaining peanut root rot (t value = −7.39, $P < 0.001$). The other variables selected by the model mainly included soil nutrient indicators, e.g., the content of soil nitrate, available potassium, total nitrogen, and total phosphorus. Soil pH revealed a significantly negative effect on disease severity (Fig. 2a, b, $P < 0.001$). Taken together, our results suggest that acid soils trend toward lower plant health and greater host plant disease.

### Soil bacterial rather than fungal communities exhibit soil acidification shifts

We obtained 1,193,050 and 2,604,284 high-quality reads for 16 S and ITS, respectively. After discarding non-bacterial and fungal operational taxonomic units (OTUs), we obtained 11,551 bacterial and 4590 fungal OTUs. 99.5% of the bacterial sequences could be assigned to 24 bacterial phyla, mainly, including Actinobacteria, Acidobacteria, Proteobacteria, Firmicutes, Chloroflexi, Gemmatimonadetes and Planctomycetes; and 96.3% of the fungal sequences could be assigned to 10 fungal phyla, mainly, including Ascomycota, Basidiomycota and Rozellomycota (Supplementary Fig. 2). In total, 501 bacterial and 352 fungal genera belonging to the above-mentioned phyla were recorded from these samples, respectively.

We first analyzed correlations between each soil property and the overall composition of bacterial and fungal communities using Mantel test analyses (Supplementary Table 6). We found that soil bacterial communities were largely associated with soil pH (Mantel test, $R^2 = 0.42$, $P < 0.001$), SOC (Mantel test, $R^2 = 0.34$, $P < 0.001$), and soil texture (Mantel test, $R^2 = 0.27$, $P < 0.001$). However, we found no significant correlations between these soil properties and fungal community composition ($P > 0.05$). Further, NMDS ordinations based on Bray-Curtis distances confirmed that bacterial communities of soil samples belonging to acidic soils (pH 4.0–5.0) clustered in distinct groups, and of samples associated with higher pH (6.0–7.0) displayed the microbial community structural along the first principal coordinate (NMDS1) (Fig. 2c, Stress = 0.079). However, soil fungal communities across soil pH ranges did not show distinctive communities, though samples belonging to higher pH (6.0–7.0) were distinguishable along NMDS1 (Fig. 2d, Stress = 0.093). Our results suggest that bacterial communities are far more influenced by changes in pH than fungal communities (Supplementary Fig. 7).

We further investigated the relationship between soil pH and microbial abundance using fluorogenic quantitative PCR. Through comparative analyses across soil samples of four pH ranges (4.0–4.5, 4.5–5.0, 5.0–6.0, 6.0–7.0), we found that soil bacterial abundances significantly increased with soil pH (Fig. 2e, $R = 0.70$, $P < 0.001$, $R^2 = 0.49$). However, we did not find any correlation between soil fungal abundances and pH (Fig. 2f, $R = -0.03$, $P > 0.05$, $R^2 = 0.001$). The lowest bacterial copy numbers were detected in soil samples with low pH (4.0–4.5), ranging from $7.8 \times 10^7$ to $5.74 \times 10^8$ copies/g soil, followed by soil pH with 4.5–5.0, ranging from $1.43 \times 10^8$ to $4.07 \times 10^9$ copies/g soil. Furthermore, α-diversity (Shannon index, Fig. 2g) and richness (Chao1 index, Supplementary Fig. 8a) of soil bacteria significantly decreased with increased soil acidity ($P < 0.01$); by contrast, levels of α-diversity and richness of fungi remained largely unchanged across different pH categories (Fig. 2h, Supplementary Fig. 8b, $P > 0.05$).

### Changes in the relative abundance of bacterial and fungal taxa under soil acidification

Phylum-level relative abundance plots revealed that two abundant (>1.5%) bacterial phyla (Chloroflexi and Planctomycetes) significantly

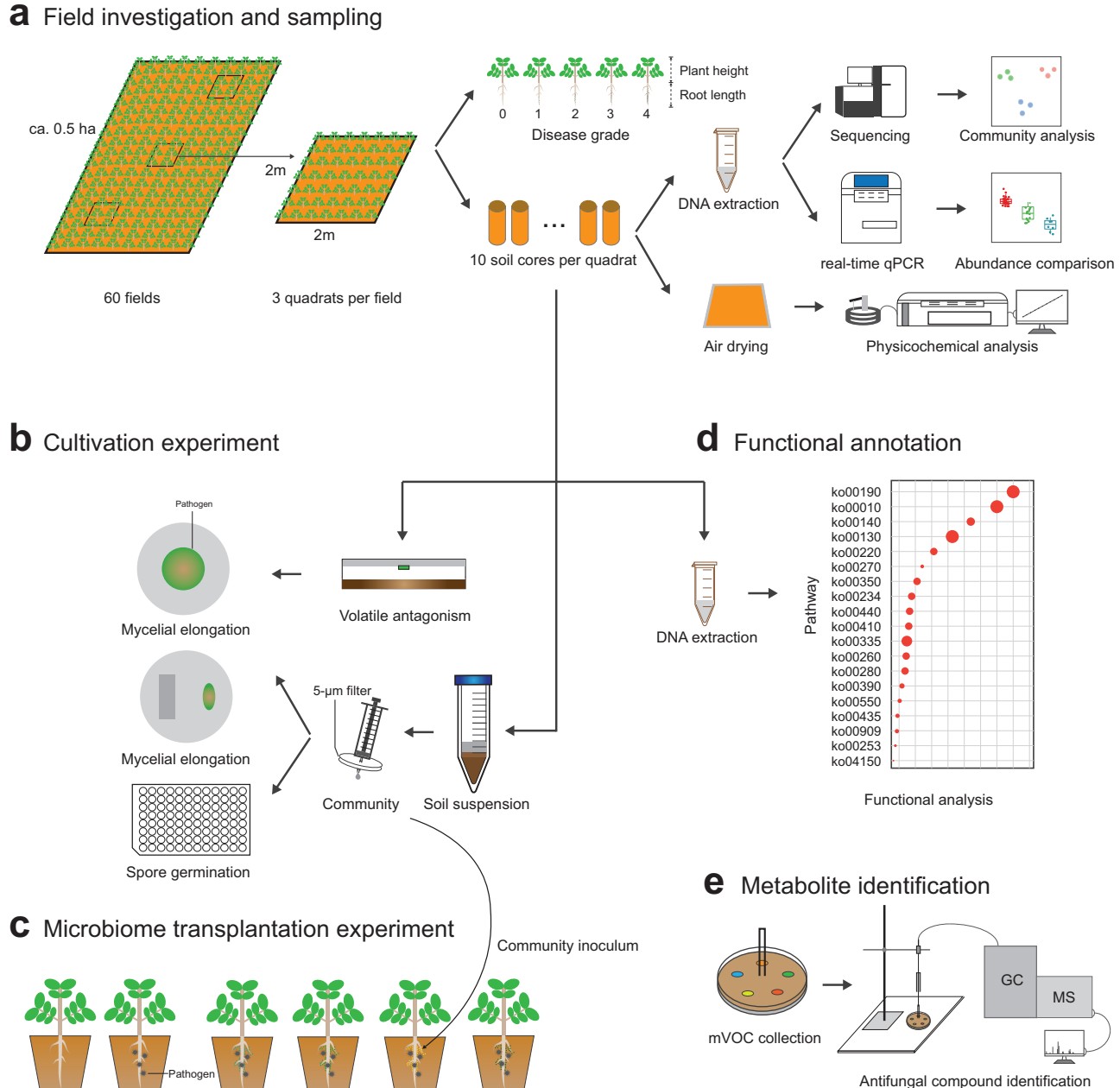

**a** Field investigation and sampling

**b** Cultivation experiment

**c** Microbiome transplantation experiment

**d** Functional annotation

**e** Metabolite identification

**Fig. 1 | An overview of the experimental workflow for this study. a** Field investigation and sampling. Three quadrats of 2 m × 2 m were selected randomly for sampling in each field. Plant disease severity was determined, and soil cores homogenized and transported to the lab for DNA extraction and soil physicochemical analysis. **b** Cultivation experiment. Fungal mycelium of the pathogen *Fusarium* sp. was exposed to soil microbial volatiles. Pathogen mycelium and spores were co-cultured with bacterial communities on agar and in suspensions to reveal the suppressive effect of bacterial communities on pathogen growth and reproduction. **c** Microbiome transplantation experiment. The bacterial suspensions obtained from soil samples were inoculated in sterilized vermiculite for

growth of peanut seedlings. Sterile deionized water was used in controls. After 30 d of spore suspension incubation, the protective role of bacterial communities towards pathogen invasion was recorded according to disease incidence. **d** Functional annotation. Metagenomic sequencing of 12 soil communities at pH of 4.0−4.5, 4.5−5.0, and 5.0−6.0 was conducted to disclose the functional profiles of the soil microbiome. **e** Metabolite identification. Volatile compounds produced by soil microbial communities were collected using glass Petri-dishes with 'chimney' lids, and identified by GC-MS for determination of volatile compounds with a potential role in pathogen suppression.

increased with acidification (Duncan's multiple comparison test, $P < 0.05$), whereas three taxa (Proteobacteria, Acidobacteria and Gemmatimonadetes) were significantly reduced as pH decreased (Fig. 3a, Duncan's multiple comparison test, $P < 0.05$). For phyla detected as significant across multiple tests (Supplementary Table 7), the direction of the response (increase or decrease in relative abundance with soil pH) was always consistent (Supplementary Fig. 7). Highly abundant classes, Planctomycetacia, Ktedonobacteria and

Chloroflexia were significantly enriched under soil acidification (from 0.8%, 10.9%, 2.4% at pH 6.0−7.0 to 2.2%, 36.7%, 3.9% at pH 4.0−4.5, respectively) (Supplementary Fig. 9, Spearman correlation, $P < 0.05$). For fungal phyla, Ascomycota dominated the soil fungal communities with abundances of 77.2%−99.3% across all samples (Supplementary Fig. 2), but the responses of the remaining fungal phyla to soil acidification had no consistent patterns across the four acidification categories (Fig. 3b, Spearman correlation, $P > 0.05$). Taken together, soil

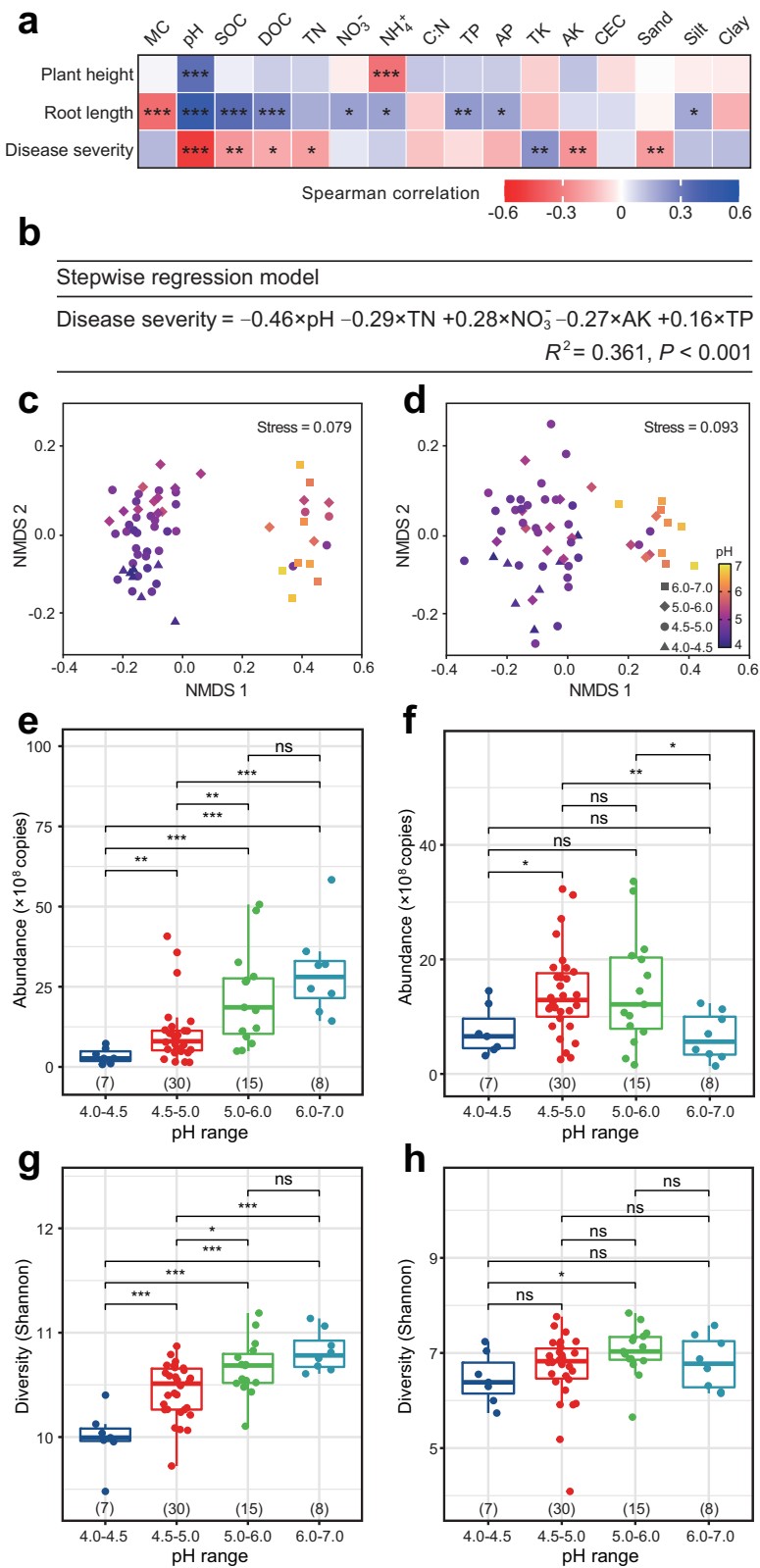

bacterial composition, rather than fungal, showed greater sensitivity to soil acidification.

To further identify bacterial families associated with soil pH, we performed a heatmap analysis at the family-level derived from the significant responses of abundant phyla to soil acidification across all samples. Of the 110 families belonging to the sensitive phyla, 80 showed strong correlation with soil acidification (Spearman

correlation, $P < 0.05$), and a phylogenetic tree was constructed to depict dominant (present in over 25% fields) microbial taxa that exhibit significant responses to soil acidification. The tree indicated a non-random phylogenetic distribution of microbial taxa in response (Fig. 3c). Of the top 30 most abundant families, Ktedonobacteraceae and JG30-KF-AS9 within Chloroflexi, Isosphaeraceae and Gemmata-ceae in Planctomycetes exhibited the highest overall preference for

**Fig. 2 | Relationship between plant agronomic indices and soil physicochemical properties, and effects of soil acidification on the soil microbiome. a** Significant Spearman's correlation coefficients were noted by asterisks. *P* values were adjusted by Benjamini–Hochberg false discovery correction (\*$P < 0.05$, \*\*$P < 0.01$, \*\*\*$P < 0.001$; $n = 180$). MC, soil moisture content; SOC, soil organic carbon; DOC, soil dissolved organic carbon; TN, total nitrogen; $NO_3^-$, nitrate; $NH_4^+$, ammonium; C:N, SOC/($NO_3^-$ + $NH_4^+$); TP, total phosphorus; AP, available phosphorus; TK, total potassium; AK, available potassium; CEC, cation exchange capacity. **b** Stepwise regression model of disease severity and soil properties. The statistical test used is *F*-test (F-statistic: 21.24), and $P < 0.001$ denotes the significance of the model and the significance of the predictor in the model ($n = 180$). NMDS ordinations of Bray Curtis distances for

the bacterial (**c**) and fungal (**d**) communities from soil samples of different pH ($n = 60$). Ranges of pH are represented by different shapes in panels (**c**) and (**d**). Copy numbers of the soil bacteria (**e**) and fungi (**f**) from soil samples of different pH ranges ($n = 60$). Shannon diversity index of the bacterial (**g**) and fungal (**h**) communities from soil samples of different pH ranges ($n = 60$). In the panels **e-h**, boxplots indicate median (box center line), 25th, 75th percentiles (box), and 5th and 95th percentiles (whiskers). Asterisks indicate significant differences as represented by the Wilcoxon rank-sum test (two-sided, \*$P < 0.05$, \*\*$P < 0.01$, \*\*\*$P < 0.001$), and "ns" means not significant difference. Numbers in brackets denote the sample size in corresponding boxplot.

soil acidification (Fig. 3d). However, the majority of the dominant family taxa within Proteobacteria exhibited significant negative associations with soil acidification (Fig. 3d, Spearman correlation, $P < 0.05$).

## Coherence of soil acidification response across taxonomic ranks to root rot

To assess the relationships between microbial composition and the observed plant diseases, we used spearman correlation analyses between significant microbial parameters with soil pH and plant disease indices. We observed that, among all the bacterial and fungal phyla, Chloroflexi was positively correlated with plant disease severity (Fig. 4a, Spearman, $R = 0.43$, $P = 0.001$, $R^2 = 0.18$), yet Proteobacteria (e.g., Alphaproteobacteria, Betaproteobacteria, Deltaproteobacteria) and Gemmatimonadetes were negatively correlated with plant disease severity (Fig. 4a, $P < 0.05$). Soil bacterial diversity (Chao1 and Shannon) was also negatively correlated with disease severity (Supplementary Fig. 10, $P < 0.05$). We used random forest modelling to further identify the main microbial predictors of plant disease severity (Fig. 4b). Consistently, the results indicated that the top five individual predictors were, in order of importance: Gemmatimonadetes with 12.5%, Alphaproteobacteria 9.9%, Chloroflexi 9.7%, Betaproteobacteria 8.2%, and Deltaproteobacteria 7.9%. No contribution of most abundant fungi to explain plant disease was observed (Fig. 4b, Supplementary Table 6).

Combining the results of correlation analysis and random forest modelling, 246 genera (Supplementary data 1) distributed in five bacterial taxa: Gemmatimonadetes, Alphaproteobacteria, Chloroflexi, Deltaproteobacteria, and Betaproteobacteria were identified across all soil samples. We then used co-occurrence networks to identify the correlation among these microbial taxa and their associations with peanut root rot. Of the 246 genera, more than 90% were clustered into three ecological modules strongly co-occurring with each other (Modules #1, #2, and #3; Supplementary Fig. 11). Within the three network modules, 13 genera in Alphaproteobacteria, 2 genera in Betaproteobacteria, 2 genera in Deltaproteobacteria, 2 genera in Gemmatimonadetes, and 4 genera (listed in Supplementary Table 8, respectively) in Chloroflexi were identified as negatively correlated with plant disease severity ($P < 0.05$), whereas 1 genus (Acidicaldus) in Alphaproteobacteria and 2 genera (Ktedonobacteraceae and 1921-3) in Chloroflexi were identified as positively correlated with plant disease severity (Supplementary Fig. 11, Supplementary Table 9, $P < 0.05$). Across 26 genera, 78.8% exhibited strong positive correlations (Supplementary Fig. 11, Supplementary Table 10, $P < 0.01$).

## Bacterial associated disease suppression and plant root protection in vitro

We further performed several in-vitro experiments (Fig. 1b) to examine the effects of soil bacterial metabolites (obtained from all soil samples) on peanut root-rot (i.e., caused by the pathogen *Fusarium* sp.). Microbial metabolites (volatiles) and bacterial inocula obtained from acidic soils (4.0–4.5) had a weaker suppressive effect on *Fusarium* sp. mycelial elongation (Fig. 5a–d) and spore germination than those

obtained from high pH soils (6.0–7.0), with suppression ability dropping by 20.6%–50.7% (Fig. 5e, f). Notably, the suppressive effects increased with increasing soil pH.

Further microscopic examination revealed the pathogenic spore germination morphology when mixed with bacterial inocula. The bacterial community from relatively high pH level of soil samples (5.0–6.0, 6.0–7.0) suppressed the conidium and chlamydospore germination, and inhibited germ tube elongation (Fig. 5e), reducing the germination rate by 43.9% and 50.7%, respectively, as compared with the acidic soils (4.0–4.5) (Fig. 5f). We further found a strong and significantly positive correlation between the relative abundance of ecological module #3 and the suppressive capacity of pathogen growth, irrespective of the mycelial elongation ($R = 0.60$, $P < 0.001$, $R^2 = 0.36$) and the spore germination ($R = 0.48$, $P < 0.001$, $R^2 = 0.23$), yet no significant correlation was detected between the relative abundance of module #1, #2, and such suppressive effect (Supplementary Fig. 12, $P > 0.05$).

We finally examined differences in protection of peanut seedling roots against pathogen infection by different soil bacterial inoculations (Figs. 1c, 5g). All bacterial inoculations from soil samples reduced *Fusarium* pathogen invasion. Interestingly, bacterial inoculations from acidic soils (4.0–5.0) owned a significantly weaker protection of root health, compared with relatively higher pH soils (6.0–7.0) (Fig. 5h, Wilcoxon rank-sum test, $P < 0.001$). Overall, the suppressive ability of bacterial communities decreased with increasing soil acidification.

## Functional capacity of the soil microbiome and identification of sulfur metabolism components

We elucidated mechanistic responses of the soil microbiome associated with soil acidification performing shotgun metagenomic sequencing of 12 soil communities at pH of 4.0–4.5, 4.5–5.0 and 5.0–6.0 (Fig. 1d). More than 97% of the sequences were high-quality after screening and filtering (Supplementary Table 11). From those sequencing reads, a non-redundant gene catalog was obtained comprising 1,668,824 genes, of which we identified 915 enriched genes (DEGs) in soils of pH 4.5–5.0, and 1074 DEGs in 5.0–6.0. We determined the functional differences in soils with pH 4.5–5.0, pH 5.0–6.0 as compared to soils of pH 4.0–4.5, and discovered a total of 58 and 28 significantly (Wilcoxon rank-sum test, $P < 0.05$) enriched KO functional categories (pathways) in pH 4.5–5.0 and pH 5.0–6.0, respectively.

To further explore the pathways in which all differential KO functional categories were affected by soil acidification, 17 KO functional categories were retained for KEGG pathway enrichment. Through this filter, it was notable that ko00920, the sulfur metabolism pathway was identified to play an important role in fungal pathogen suppression by the microbiome, as it was enriched in all soils of relatively higher pH (Fig. 6a, Wilcoxon rank-sum test, $P < 0.001$). Concomitantly, effector genes and enzymes, including *soxA, B, C, X, Z, Y, fccB, dmdA, cysP* and APA1, were significantly increased in relative abundances in the microbiome of all soils with higher pH compared to that of lower pH (Fig. 6b, two-sided *t*-test, $P < 0.05$). These genes and enzymes participate in the synthesis of sulfur compounds, such as

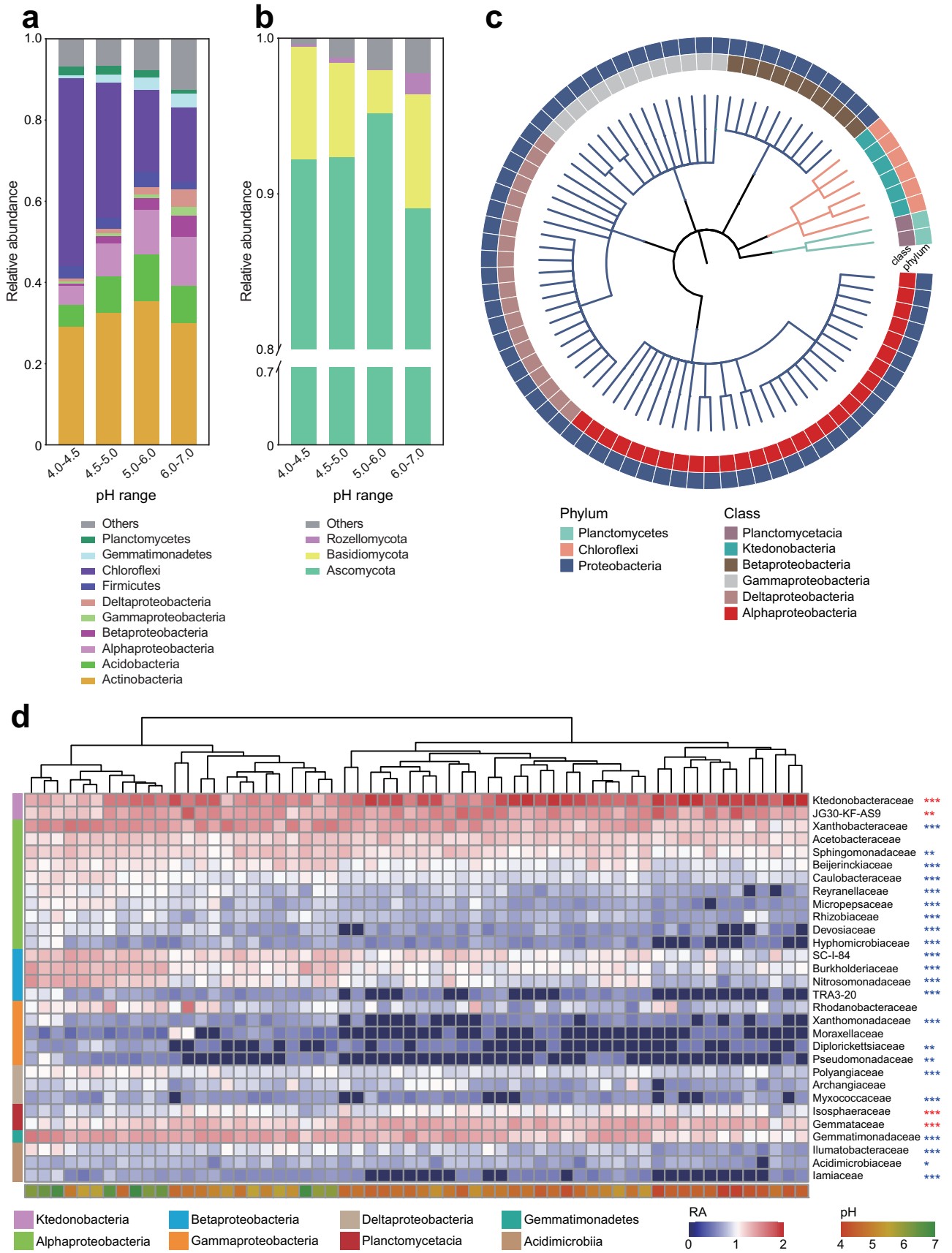

**Fig. 3 | Responses of bacterial and fungal communities to soil acidification.** Relative abundances of bacterial (**a**) and fungal (**b**) communities at the phyla level from soil samples of different pH ranges. **c** Phylogenetic tree depicting microbial taxa with significant responses to soil acidification. **d** Heatmap of the top-30 abundant bacterial families across soil samples of different pH. Hierarchical clustering analysis was performed using the neighbor-joining method. The left gradient color key represents the relative abundance (RA) of bacterial families, whereas the right gradient color key represents soil pH. Asterisks indicate significant correlations (*$P < 0.05$, **$P < 0.01$, ***$P < 0.001$, $n = 60$) between RA and pH across samples by Spearman's method (Blue, positive; red, negative).

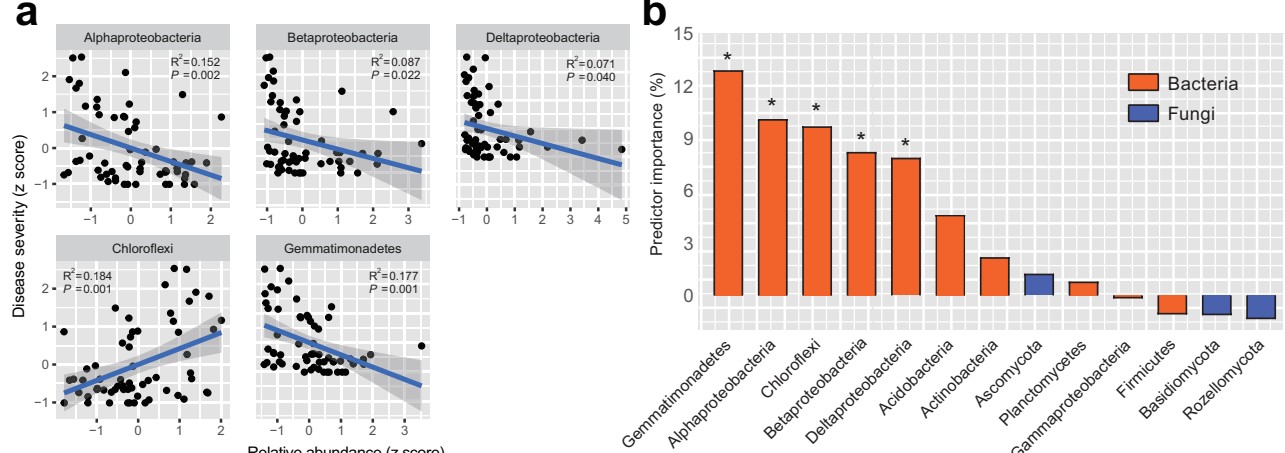

**Fig. 4 | Microbial taxonomic responses as bioindicators of disease severity.** **a** Ordinary least squares (OLS) linear regression between the disease severity index (DSI) and the relative abundances of bacterial phyla ($n = 60$). Only significant fitted lines between DSI and relative abundance of microbial phyla are displayed on the graphs. The blue fitted lines are regression lines from OLS regression, and the shaded areas indicate 95% confidence interval of the fit. The statistical test used is $F$-test based on two-sided tests, and $P < 0.05$ denotes the overall significance of the regression model. **b** Mean predictor importance (increased mean square error, % MSE) of bacterial and fungal phyla on disease severity index based on random forest modelling. Asterisk indicates significant random forest importance.

DMSO, DMSP, and dimethyl sulfone. The remaining differential KO functional categories, such as two-component system [PATH:ko02020], peptidoglycan biosynthesis [PATH:ko00550], bacterial secretion system [PATH:ko03070], and glycerophospholipid metabolism [PATH:ko00564] were mainly involved in components of microbial carbohydrate and energy metabolism. These results suggest that soil acidification may reduce gene expression of bacterial communities involving in fungal pathogen suppression, further supporting previous results that reveal responses of bacterial community composition and reduced pathogen inhibition along the soil acidification gradient.

We next sought to determine whether the pH-induced shifts in our soil metagenomics, specifically the reduced expression of sulfur metabolism genes, were correlated with changes in the metabolism of specific antifungal substances, and performed untargeted metabolomics on bacterial communities by using gas chromatography–mass spectrometry (GC-MS) at pH of 4.0–4.5, 4.5–5.0, and 5.0–6.0 (Fig. 1e). Through comparative analyses across different soils, we identified many identifiable pH-induced bacterial metabolites ($n = 28$), including a variety of alkanes, benzenes and sulfur compounds (Supplementary Table 12). Interestingly, the most significantly enriched metabolite was dimethyl disulfide (DMDS), 45.0–45.9 folds more abundant in high pH soils than that of lower pH (Wilcoxon rank-sum test, $P < 0.001$), again, further suggesting an important role of bacterial sulfur metabolism in potential pathogen suppression that correlates with shifts in soil pH.

## Discussion

Our study provides solid evidence that soil acidification in arable fields can have an important impact in promoting plant disease severity caused by plant pathogens, specifically via the inhibition of soil microbiome fungal pathogen suppression capabilities. In particular, soil acidification leads to stronger changes in bacterial communities than in soil fungi, resulting in a declined capacity of soils to suppress

*Fusarium* root rot. We further used shotgun metagenomics and untargeted metabolomics, and showed that a small subset of functional pathways and microbial traits related to sulfur metabolism decline in acidic soils. Specifically, we found significant down regulation of sulfur compound synthesis genes and decline in a known antifungal bacterial metabolite, and perhaps most importantly, these declines in microbiome sulfur-associated functions are correlated with plant disease severity. Our results suggest that under soil acidification of arable fields, synergistic effects of soil physicochemical properties and microbial structure and function can determine plant health status.

The role of pH as a key environmental factor influencing microbial communities has been described by studies at the continental and nationwide scale, and also with paired comparisons in numerous individual sites[24–27], yet how soil pH affects the capacity of the soil microbiome to regulate plant disease has not been previously elucidated. Apart from the contribution of atmospheric sulfur dioxide and nitrogen oxides produced by fossil-fuel combustion, intensive management in exceedingly limited agriculture land area, including increased application of ammonium fertilizer (rather than organic fertilizers) and the subsequent removal of base cations by plants, contributes substantially to soil acidification in China[25,28,29]. The process of soil acidification induces the evolution and re-assembly of soil bacterial communities[15,30–32]. For instance, Delgado-Baquerizo et al.[33] and Guerra et al.[34] at continental scales, and Malik et al.[35] and Tripathi et al.[30] spanning local to global scales and short- to long-term successional trajectories, indicated assembly of specific soil bacterial communities via niche-based exclusion in acidic pH environments. By contrast, variations in other edaphic characteristics (e.g., AP, AK, CEC and sand content) were poor predictors of bacterial community organization, suggesting that soil nutritional status has relatively less impact on top-level assembly of soil bacterial communities. Meanwhile, the declines of soil bacterial richness and diversity exacerbated

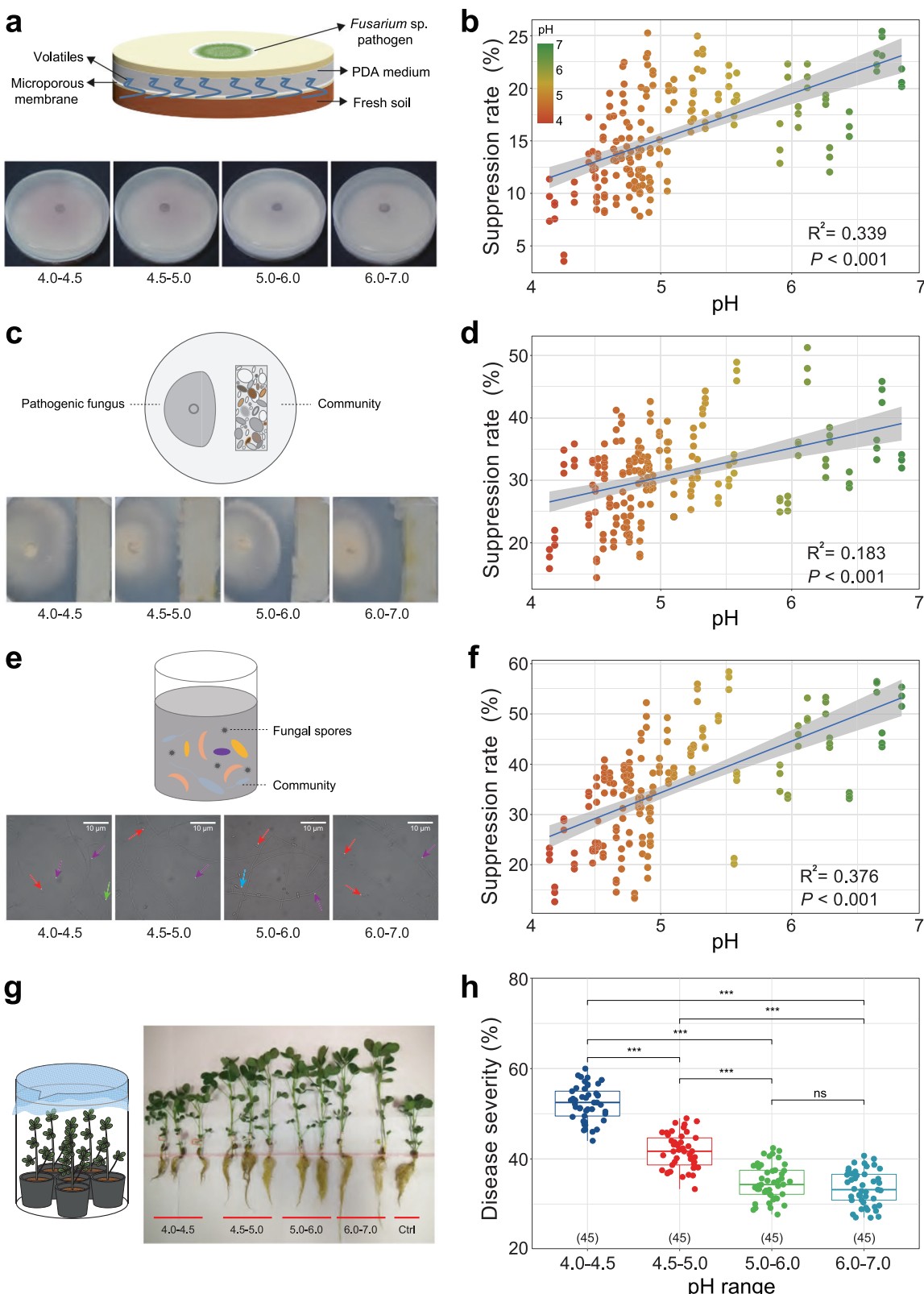

by soil acidification can cause significant changes in the bacterial community. Recently, Luan et al.[36] mechanistically integrated pH into the metabolic theory of ecology to predict soil bacterial diversity, and found that the integrated model reliably and accurately predicted the patterns of bacterial diversity across soil pH gradients. Acidic conditions can disrupt the energy metabolism of soil bacterial taxa by reducing their proton gradients, which can impede the survival of bacterial taxa that have high energy demands or low tolerance to acidic conditions[24,37,38]. Soil fungi with nuclei surrounded by nuclear membranes rather than single-celled bacteria, show strong tolerance to external stress of extracellular pH[39,40]. The adaptations allow fungi to withstand the stress imposed by low pH conditions and enable them to occupy ecological niches that may be less suitable for many bacterial groups[41,42].

**Fig. 5 | Pathogen-suppressive ability of bacterial communities extracted from soil samples of different pH ranges and their protection against *Fusarium* root rot. a, b** Experimental set-up and results of response of mycelial growth of pathogenic fungus *Fusarium* sp. to soil volatiles. **c, d** Experimental set-up and results of response of the pathogenic fungus *Fusarium* sp. growth to soil bacterium suspension. **e, f** Experimental set-up and results of response of spore germination rate of *Fusarium* sp. when co-cultured with soil bacterium suspension. The red arrows in panel **e** indicate the conidia; the green arrow indicates the chlamydospore; the blue arrow indicates the germ tubes; and the purple arrows indicate the hyphae. In **b, d** and **f**, the blue fitted lines are regression lines from OLS linear regression between the suppression rate and soil pH ($n = 180$), and the shaded areas

indicate 95% confidence interval of the fit. The statistical test used is $F$-test based on two-sided tests, and $P < 0.001$ denote the overall significance of the regression model. The color key represents soil pH. **g** Experimental set-up and results of response of the protective efficacy of the inoculated bacterial suspension to intrusion of *Fusarium* sp. **h** Disease severity of peanut root after colonization of bacterial suspensions obtained from soil of different pH. Boxplots indicate median (box center line), 25th, 75th percentiles (box), and 5th and 95th percentiles (whiskers). Asterisks indicate significant differences as represented by the Wilcoxon test (two-sided, *$P < 0.05$, **$P < 0.01$, ***$P < 0.001$), and "ns" means not significant difference. Numbers in brackets denote the sample size in corresponding boxplot ($n = 45$ for each pH category).

Compared with fungal communities, soil bacteria play a more vital role in mediating the severity of soil-borne disease. On one hand, higher diversity and richness of soil bacterial communities induced a stronger suppression on soil-borne pathogens, while the effects of diversity and richness of soil fungal communities on disease severity are not obvious[43,44]. In acidic soils of this study, resident soil bacterial community with less diversity may struggle to resist invasion of fungal pathogens, thereby compromising the maintenance of a robust microbiota homeostasis against soil-borne pathogen infections[45,46]. On the other hand, key bacterial groups (e.g., Proteobacteria) which play a significant role in disease suppression[47] were suppressed at low soil pH levels in our study. This was supported by random forest modelling revealing that five bacterial phyla have prominent effects on disease occurrence, yet no effects were identified in fungal taxa. For instance, Alpha-, Beta-, and Delta-proteobacteria were negatively correlated with plant disease severity (Fig. 4). The community assembly patterns of these proteobacterial taxa have been shown to shift with soil acidity[25]. It is important to note that these proteobacterial classes may rely on synergistic interactions to exert their pathogen-suppressive effects[43,48]. Our network analysis showed that approximately 90% of the interactions among Proteobacteria exhibit positive associations (Supplementary Fig. 11), indicating their mutualistic cooperation in achieving inhibitory effects against pathogen infections.

Meanwhile, soil acidity is directly related with pathogenicity, as *Fusarium* can be more aggressive at low pH[49]. To further investigate the causal relationship between soil bacterial communities and soil-borne disease prevention, we conducted multiple inoculant experiments by extracting soil bacterial inocula and culturing them directly with the hyphae and spores of the fungal pathogen *Fusarium*. Our results confirmed that bacterial communities from acidified soils lost the capacity to inhibit fungal pathogen growth, resulting in reduced capacity to suppress pathogen growth and ultimately efficacy. This is consistent with our sequencing results that showed acidified soils harbor higher abundances of pathogenic fungi, *Fusarium solani* (Supplementary Fig. 13). We then conducted additional experiments wherein we inoculated soil suspensions into the substrate used to cultivate peanut seedlings. We found that microbial inocula coming from more acid soils had a lower capacity to protect plant roots than those coming from near-neutral pH soils. Collectively, soil acidification may disrupt the capacity of bacteria to suppress pathogen growth and reproduction, which can result in more severe occurrence of plant diseases[50,51].

We then aimed to understand the mechanisms explaining the capacity of the soil microbiome to protect plants from disease across a range of soil pH. We showed that downregulation of sulfur metabolism pathways in acidified soils accompanied the declined microbiome suppression of fungal pathogen. Genes encoding for sulfite oxidase (*soxA* and *soxB*), and sulfane dehydrogenase (*soxC*) were significantly downregulated in acidified soils. This could be associated with the depletion of bacterial taxa (e.g., keystone taxa *Dongia* and *Hyphomicrobium* in the Alphaproteobacteria, Supplementary Table 8) that mediate sulfur cycling, thus indicating that the loss of important bacteria can impact sulfur metabolism pathways[52,53]. *Dongia* and

*Hyphomicrobium*, which are depleted in acidic soils, have been reported to metabolize sulfate or thiosulfate into sulfur compounds[52,54]. In our study, we show that soil acidification disrupted the emission of antifungal substances, and inhibited the synthesis of sulfur compounds (e.g., DMSO and DMSP), which have been predicted to decline under soil acidification[55]. Previous studies have shown that DMSO has a strong inhibitory effect on pathogenic fungi via direct cell membrane destruction or by indirectly disorganizing the functioning of cellular antioxidation systems in pathogenic fungi[4,46,51]. Potential triggers of the sulfur signaling pathway in the soil-microbe-plant system need to be further elucidated, but we propose that soil acidification has strong effects on the organization of soil microbial consortia, particularly bacteria, which then impacts the functional capacity to confront pathogens via a reduction of sulfur-associated bacterial genes and biochemicals that play important roles in fungal pathogen suppression[50,56].

Finally, we posit that it is fundamental to consider the broader context of crop disease management. Several factors, such as the absence of crop rotation practices, the use of susceptible cultivars, the excessive application of pesticides and fertilizers, and the abandonment of indigenous seed microbiomes have been identified as contributors to the greater occurrence of soil-borne diseases in agricultural ecosystems[57,58]. Our results revealed that the native soil microbiome plays an essential role in regulating the defense capacity of plants against fungal pathogens. Changes in the soil microbiome associated with acidification linked with agricultural management can have important consequences for plant disease severity[59,60], and we provided experimental and mechanistic explanations on why this is the case. Our results further highlight the role of soil-borne microbial legacies in shaping plant health for food production[9]. The complex interactions between plants and soil microbiome are still being tackled, with important work now highlighting microbial-driven defense mechanisms against pathogen infections[61]. Further investigations considering the influence of the soil microbiome and its legacy on pathogen infection are needed to better understand the future of food production under global environmental change.

Our study revealed the major contribution of soil acidification to explain a complex microbiome-mediated system that can control plant disease severity. Soil acidification had important negative effects on bacterial abundance and diversity, whereas it induced minor changes in the soil fungal community. Further inoculant experimental assays provided evidence of the fundamental role of bacterial communities in suppressing plant disease severity from *Fusarium* in arable soils; a positive disease-suppressive effect which was reduced when using microbes from acid soils. Ultimately, the bacterial taxa associated with low pH soils concomitantly have a low functional capacity to suppress plant pathogens, thus contributing to more severe plant diseases. Overall, our findings advance our understanding of the effect of soil acidification on plant health regulated via changes in soil bacterial communities, and will serve to guide future soil microbiome manipulations that aim to promote plant health and sustainable agriculture.

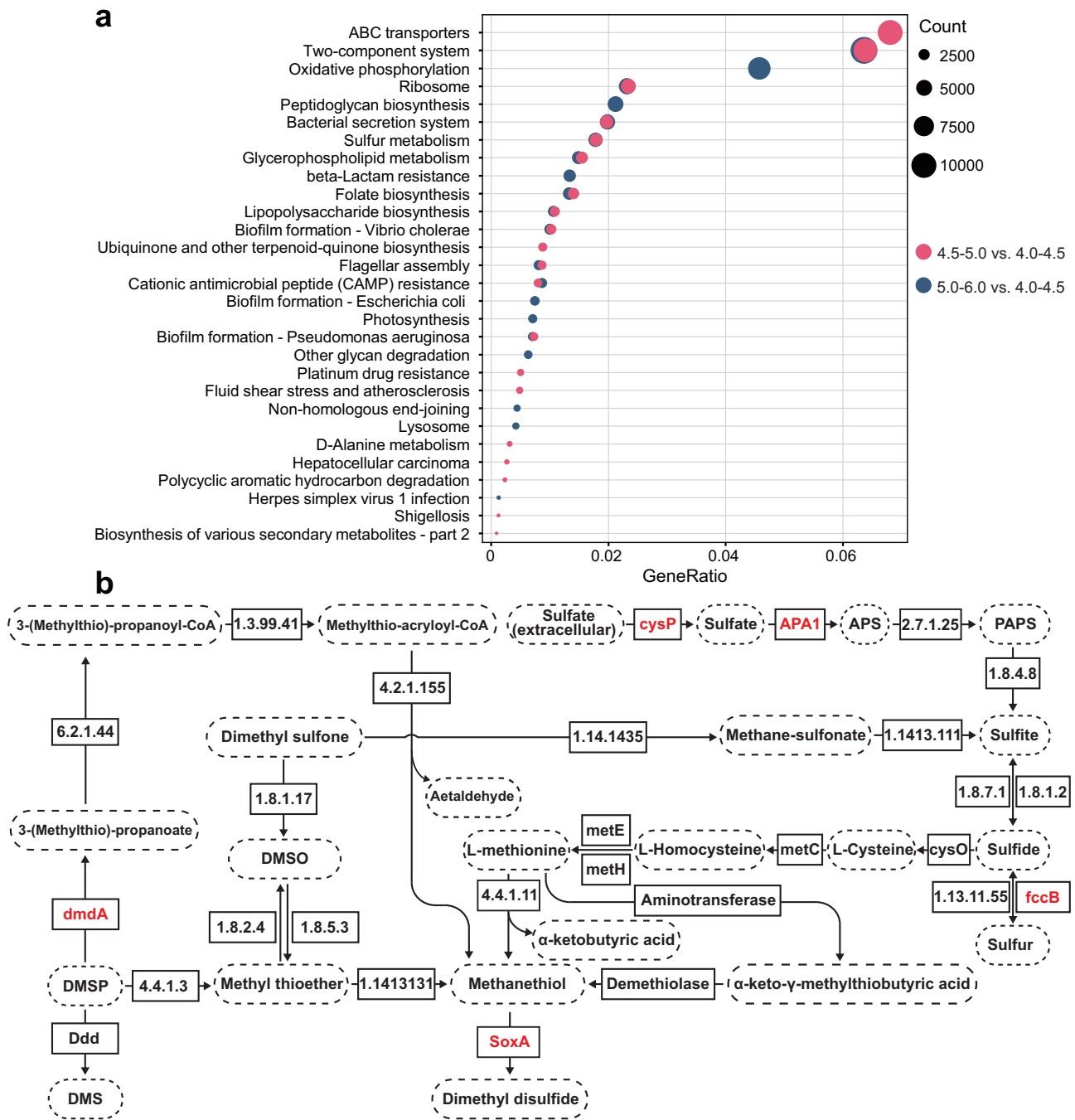

**Fig. 6 | Mechanistic responses of the soil microbiome associated with soil acidification.** Main differentially abundant KO functional categories in KEGG pathway enrichment (**a**), and key metabolic pathways of sulfur compounds (**b**). **a** Texts on the left indicate level 3 KEGG functional categories. **b** Texts in the solid line box represent genes or enzymes mediating corresponding metabolic process. Texts in the dashed line oval represent metabolites. Red texts in the solid line box indicate down-regulated genes under acidified soil conditions.

## Methods

### Study site

A total of sixty crop fields were randomly sampled across southeast China (Fig. S1) (28.10–28.90°N, 115.00-116.96°E). Soils correspond with quaternary red clays [Udic Ferrosol, FAO (1998)] with zonal distribution. These soils are characterized by support low concentrations of organic matters and fertility. Peanut (*Arachis hypogaea* L.) is widely planted across these soil regions. Fields were planted in early April with a homologous peanut cultivar (Ganhua) and sampled in late July 2018, shortly before sprout of root disease.

### Plant and soil surveys

For our field survey, three quadrats (replicates, 2 m × 2 m) were randomly arranged in each field (about 0.5 ha). We sampled (using a square frame) paired samples of plants and soils in each quadrat (60 sites x 3 quadrats; $n$ = 180). In detail, 50 plants of each quadrat were randomly excavated, and 9,000 plants (60 fields x 3 quadrats x 50 plants) were surveyed to determine disease severity. The root rot symptoms were assessed by scoring as follows: 0, healthy plant without any root rot symptoms (no infection); 1, mild infection with root rot symptoms (1–25% of the root surface area infected); 2, moderate infection with root rot symptoms (26–50% of the root surface area

infected); 3, serious infection with root rot symptoms (51–75% of the root surface area infected); and 4, severe infection with root rot symptoms (76–99% of the root surface area infected or a dead plant). The degree of root rot was expressed as the disease severity index (DSI), which was calculated using the formula $DSI(\%) = \sum_{i=1}^{4} \frac{n \times i}{N \times 4} \times 100$. Also, the plant height and root length were determined immediately in the field (Fig. 1). For soil sampling, 10 soil cores (5-cm diameter and 15-cm depth) in the inter-row of each quadrat after removing and discarding surface materials were collected to pool and homogenize. This generated three composite samples per field, and 180 soil samples were kept at 4 °C during transportation. Upon arrival in the lab, soil samples were sieved through a 2-mm mesh to remove visible residues (e.g., rocks, roots and organic debris), and subdivided into three portions. One portion was air-dried and further sieved through a 0.25-mm sieve for determination of soil physicochemical properties. Another portion was frozen at −40 °C until DNA extraction for microbial community, abundance and functional analyses. The last portion was stored at 4 °C for conducting microcosm experiments (Fig. 1).

## Soil physicochemical characteristics

Soil physicochemical properties were determined according to handbook of soil analysis[62]. Soil moisture content (MC) was determined by using the gravimetric method after a 16-h desiccation at 105 °C. Soil pH was measured with a glass electrode in a soil-to-water ratio of 1:2.5 (w/v). The content of soil organic carbon (SOC) was determined by titrating against 0.5 M ferrous iron solution after the air-dried soil had been digested with 0.8 M $K_2Cr_2O_4$ and concentrated $H_2SO_4$ (v/v, 1:1) at 150 °C for 30 min by the potassium dichromate oxidation method. Dissolved organic carbon (DOC) was extracted by incubating fresh soil (equivalent to 10 g dry mass soil) with 50 mL of deionized water for 30 min, followed by shaking (end-over-end) at 280 rpm for 30 min at 25 °C. Then the samples were transferred to a centrifuge tube, and centrifuged at 450x $g$ for 20 min at 4 °C. Subsequently, the supernatants were filtered through a 0.45-μm cellulose nitrate membrane before quantifying the DOC content by using a total organic carbon (TOC) analyzer (Multi N/C 3100, Germany). Total nitrogen (TN) was measured by the Kjeldahl digestion method. Briefly, the soil sample was heated and boiled with concentrated $H_2SO_4$. The solution was then absorbed by 2% boric acid solution and titrated against 0.1 M sulfuric acid. Ammonium ($NH_4^+$) and nitrate ($NO_3^-$) were extracted by using 1 M potassium chloride, and then determined by the UV spectro-photometry method. Total phosphorus (TP) and available phosphorus (AP) were determined by using the molybdenum-blue method with an atomic absorption spectrophotometer, after the soil had been digested with concentrated $HF$-$HClO_4$ (v/v, 2:1) or extracted using 0.5 M sodium bicarbonate, respectively. Total potassium (TK) and available potassium (AK) were determined by the flame emission spectrometry method, after the soil had been digested in concentrated $HF$-$HClO_4$ (v/v, 2:1) or extracted with 1 M ammonium acetate, respectively. To examine the effect of soil buffering capacity on changes in soil acidity, soil cation exchange capacity (CEC) was also determined through the 1 M ammonium acetate method by using the air-dried soil samples. Soil texture mainly included the proportional contents of clay, silt and gravel in soil, which were determined by the sieve-pipette method. Overall, 180 soil samples were determined.

## Soil DNA extraction, PCR amplification, and Illumina sequencing

After identifying soil pH as a dominant soil factor mediating plant disease severity, the sampled soils were classified into four categories based on soil pH (i.e., pH 4.0–4.5, 4.5–5.0, 5.0–6.0, and 6.0–7.0). Soil samples falling into the corresponding pH category were adopted as independent replicates to compare the patterns of soil microbial community among the four pH categories. Consequently, genomic DNA was extracted from 0.35 g of each soil sample using the FastDNA

SPIN kit for soil (MP Biomedicals, Santa Ana, USA) according to the manufacturer's instructions. For microbiome sequencing, the extracted DNA from three soil samples per field was pooled at the site level, resulting in 60 composite DNA samples[63]. Again, this approach was chosen because these fields have been consistently planted with peanuts for more than five years, leading to homogenization of the soil properties. The final concentration and quality of DNA were assessed based on the absorbance ratios at 260/280 nm and 260/230 nm by using NanoDrop 2000 spectrophotometer (Thermo Scientific, Wilmington, USA).

The V3-V4 regions of bacterial 16 S rRNA genes were amplified by using the primer pairs 338 F/806 R. The thermocycling conditions involved 3 min at 95 °C and then subjected to 30 cycles of denaturation at 95 °C for 30 s, annealing at 55 °C for 30 s, followed by 72 °C for 45 s, and a final extension at 72 °C for 10 min. The primer pairs ITS3/ITS4 were used for amplification of the fungal internal transcribed spacer (ITS) region. The amplification conditions included 3 min at 95 °C, and then 35 cycles of denaturation at 94 °C for 1 min, annealing at 51 °C for 1 min, followed by 72 °C for 1 min and a final extension at 72 °C for 10 min. Details regarding the primers can be found in Supplementary Table 1. For each sample, the primers utilized in the amplification reactions included unique error-correcting barcodes of variable length. The amplification reactions were carried out in a total volume of 20 μL, which consisted of 4 μL of 5× FastPfu Buffer, 2 μL of 2.5 mM dNTPs, 0.8 μL of both the forward and reverse primers, 10 ng of template DNA, and 0.4 μL of FastPfu DNA Polymerase (TransGen Biotech., China). To minimize potential biases introduced by individual PCR reactions, each sample was amplified in triplicate, and the resulting amplicons were pooled together. All PCRs were performed using the ABI GeneAmp® 9700 Thermal Cycler (Thermo Fisher Scientific, USA). The PCR products were assessed on 2.0% agarose gel with ethidium bromide staining to confirm the correct amplicon length. Subsequently, the gel-purification method was employed using the AxyPrep DNA Gel Extraction Kit (Axygen Biosciences, USA) to obtain purified amplicons. The purified amplicons were then combined in equimolar concentrations and subjected to paired-end sequencing (2 × 300 bp) using an Illumina MiSeq PE 250 sequencer (Illumina, USA) at Shanghai Personal Biotechnology Co., Ltd (Shanghai, China), following standard protocols.

## Determination of soil microbial abundances by quantitative real-time PCR

Quantitative real-time PCR (qPCR) was used for the quantification of total bacteria and fungi in the soil samples. Briefly, abundances of bacteria and fungi were quantified using primers Eub338F/Eub518R and ITS1f/5.8 s, respectively (Supplementary Table 2). Standard curves were generated by using 10-fold serial dilutions of a plasmid containing a full-length copy of the 16 S rRNA gene from *Escherichia coli* and the 18 S rRNA gene from *Saccharomyces cerevisiae*. Assays were carried out on ABI StepOne Plus™ Real-Time PCR System (ABI, USA) in a 20 mL reaction volume containing 10.0 mL of SYBR® *Premix Ex Taq*™ (Takara Bio Inc., Kusatsu, Japan), 0.8 mL of each primer (10 mM), 0.4 mL of ROX Reference Dye II (50 ×), 2 mL of template DNA, and 6 mL of sterile water. The qPCR conditions were 30 s at 95 °C, followed by 40 cycles of 95 °C for 5 s and 60 °C for 34 s[64]. Template DNA in negative controls was replaced with nuclease-free water (Qiagen, Valencia, USA). Melting curve and gel electrophoresis were performed to confirm amplification specificity. The approximate length of the targeted amplicon region was 200 bp and 300 bp for bacteria and fungi, respectively. Gene copy numbers of the target group for each reaction were calculated from the standard curves. Each essay was performed in three replicates, and the results were expressed as $\log_{10}$ values (target copy number $g^{-1}$ soil) for further statistical analysis[65].

## Response of the mycelial growth of soil pathogen to microbiome in vitro

To assess the growth responses of root rot pathogen to soils of different pH, plate cultivation experiments were performed in sterilized Petri dishes (9-cm diameter) containing top and bottom growth areas. *Fusarium* sp. ACCC 36194, a model root rot pathogen isolated from a diseased peanut root, was used[66]. Before the cultivation experiments, a fungal plug (6-mm diameter) was aseptically transferred to potato dextrose broth (PDB) agar (Supplementary Table 3) and placed in a biochemical incubator at 28 °C for 4 d in the dark to activate the growth vigor of the pathogenic fungi. Meantime, soil stored at 4 °C from each field was moistened to 40% of the water-holding capacity at 28 °C for 3 d in the dark to allow for the microbial activity to stabilize.

Homogenized soil (equivalent to 20 g dry mass soil) was spread evenly at the bottom of the Petri dish at 28 °C for 1 d. The lid of the Petri dish contained 15 mL of half-strength sterile PDA medium, with 50 mg L$^{-1}$ streptomycin added to inhibit the growth of bacteria. The lid was then inoculated in the center with a sterile PDA plug containing fungal hyphae and conditioned at 28 °C for 24 h. Then, the lid was inverted and attached to the bottom before placing a piece of a sterilized 0.22-µm microporous membrane on the bottom soil to avoid contamination onto the top compartment with a tight Parafilm seal, and the dishes were incubated at 28 °C for 3 d. In this manner, the tested fungus was exposed (without direct contact) to the volatiles produced by the soil microorganisms in the bottom compartment. After the incubation, the elongation of fungal hyphae was determined and compared with that in control plates (fungi exposed to the microporous membrane without the soil). Radial growth of *Fusarium* sp. was determined by calculating the distance (in cm) from four equidistant points from the center of the plug to the colony edge. The soil samples from each quadrat across all surveyed fields were subjected to five technical replicates, and the resulting data were averaged for further analysis.

To further assess the responses of fungal pathogen to the soil bacterial community, a pairwise antagonistic experiment was conducted. Bacterial suspensions were first prepared. Briefly, 5 g (dry weight equivalent) of soil was placed in 45 mL of phosphate buffer (KH$_2$PO$_4$, 1 g/L, pH = 6.5), and mixed on a rotary shaker at 4 °C and 150 rpm for 1.5 h. The suspension was then sonicated for 1 min at 47 kHz, twice, and mixed again for 0.5 h[67], before filtering through a 5-µm membrane to remove a large proportion of fungal propagules[68]. Then, 50 µL of the suspension was inoculated evenly on a piece of sterilized cellulose filter (1 cm × 3 cm) placed on a side of NA medium (Supplementary Table 3). After cultivation in the dark for 72 h, a PDA plug (6-mm diameter) containing *Fusarium* sp. fungal hyphae was placed on the medium, 2 cm away from the inoculated suspensions. The plate was then sealed with Parafilm and incubated at 28 °C. Mycelial growth morphology was captured after 72 h by using a stereomicroscope (Leica LED2500, Schweiz). Plates with an equal volume of sterile deionized water instead of the bacterial suspension were used as a control. Five technical replicates were conducted for soil suspensions from each quadrat as well as the control, and the resulting data were separately averaged for subsequent analysis.

## Response of pathogen spore germination to soil microbiome

To determine the response of the germination of *Fusarium* sp. to soil bacterial suspension in detail, the spore suspension of root rot pathogen, *Fusarium* sp. ACCC 36194, was first prepared[66]. Briefly, a fungal agar block was carved from a PDA plate with 7-d old mycelium and placed in a center of fresh PDA plate. After 3-d incubation at 28 °C in the dark, 10 mL of sterile deionized water was added. The spores and mycelium were manually shaken for 30 s, and then the mycelial covered surface was scraped off with a sterile glass rod. The spores were then separated from mycelial fragments and agar fragments by filtering the mixture through sterile glass wool. The filtrate was centrifuged

at 3000 x g for 10 min and cell density adjusted to a final concentration of $1 \times 10^4$ conidia mL$^{-1}$ in sterile deionized water, and preserved as a spore suspension at 4 °C until use.

The effect of bacteria community obtained from sampled soils on spore germination was determined as follows. For the experiment, 270 µL of bacterial suspension (prepared as described) and 30 µL of spore suspension were mixed in a well of a sterile 96-well plate[66]. In blank control experiments, an equivalent volume of sterile deionized water was used instead of the bacterial suspension. After 12-h incubation at 28 °C, spore germination was determined microscopically (Nikon Eclipse Ci-L, Tokyo, Japan), by counting 10 fields (40 × magnification) per well. Once the length of the germ tube was as long as that of the spore, it was considered germinated. Soil suspensions from each quadrat, as well as the control, underwent five technical replicates, and the resulting data were averaged independently for subsequent analysis.

## Microbiome defense of plant seedling against pathogen invasion

To better assess the effects of soil bacterial communities from different pH on pathogens colonizing in the plant roots, peanut seedling cultivation experiment in pots under sterile conditions was established (Fig. 1c). We chose 12 soil samples collected from the fields that 3 from pH 4.0–4.5, 3 from pH 4.5–5.0, 3 from pH 5.0–6.0 and 3 from 6.0–7.0, with the following selection criteria: 1) different pH values; 2) yet similar other physicochemical properties. In total, 36 bacterial suspensions (12 samples × 3 triplicates) were prepared, and sterile deionized water instead was used as a control.

Seedling cultivation was performed in a setup under sterile conditions[69]. Firstly, peanut seeds were surface-disinfected in 75% ethanol for 10 min, followed by three rinses in sterile deionized water, immersion in 0.1% HgCl$_2$ for 3 min and five final washes in sterile deionized water. Then, seeds were aseptically transferred onto moist filter paper in 180-mm diameter Petri dishes, and held in a biochemical incubator at 25 °C for 5 d until they pre-germinated. Contaminated seeds infected with ambient fungi were discarded and not used in the subsequent procedure. Seedlings of similar size (~ 2 cm height) were selected and individually planted in 100-mL plastic pots filled with sterilized vermiculite. Each pot was supplemented with 25 mL of sterile Hoagland's nutrient solution at a 1/4 strength, ensuring that only one seedling was cultivated per pot. The used vermiculite was autoclaved twice at 121 °C for 20 min with 24 h in between. Six 100-mL pots for each bacterial suspension sample were then placed in a 5-L beaker (Fig. 5g), which was covered with four layers of sterile gauze to prevent microbial contamination, and placed in a growth chamber with the following day/night cycle: 16 h 30 ± 2 °C /8 h 25 ± 2 °C and 75% relative humidity. The Hoagland's nutrient solution was replenished every 3 days.

On the 8th day, 15 mL of bacterial suspensions or sterile deionized water were added to the 2-cm radial area around the peanut roots at the intersecting surface of the air and the vermiculite. After incubation of 1 d, 5 mL of the spore suspension of root rot pathogen was added once per day for the next three days to where bacterial suspensions or sterile water had been added. The spore suspensions were prepared as above, with cell density adjusted to a final concentration of $1 \times 10^7$ conidia mL$^{-1}$. The incidence grade of root disease of six seedlings in a 5-L beaker was determined to calculate the disease severity index (DSI), as the methods described above, after incubation at 28 °C for 30 d. The cultivation was repeated for five times. Overall, 1110 peanut plants ((36 bacterial suspensions + 1 control) × 6 plants × 5 replicates) were collected for determination of disease infection.

## Illumina Miseq data processing

Raw sequences of soil samples were split using QIIME pipeline (http://qiime.org/) and paired-end sequences were merged by using FLASH[70]. Briefly, the forward and reverse primer sequences were trimmed after

the raw reads were assigned to samples according to their unique barcodes. Paired sequences were then assembled via paired-end reads through FLASH. Quality trimming was conducted using the following criteria: (1) Sequence reads were truncated when three consecutive low-quality bases were encountered, and the resulting sequences were re-evaluated for their length, and (2) Any reads containing ambiguous bases were removed from the analysis. Chimeric reads were distinguished and removed by using VSEARCH (v1.4.0)[71]. The remaining sequences underwent an additional screening step using HMM-FRAME to detect and remove any sequences with frame shifts. Given that we aimed to identify broad taxonomic groups of soil microbial communities across samples of a pH gradient, the sequences retained for each sample were then clustered and assigned to OTUs at a clustering threshold of pairwise identity of 97% via UPARSE pipeline. Finally, each OTU was taxonomically annotated by SILVA database (v123) for bacteria and the UNITE database (v7.1) for fungi[72]. Alpha diversity values and Bray-Curtis dissimilarity for beta-diversity analyses were then calculated for soil bacterial and fungal communities. To ensure fair comparisons, all samples were rarified to equal sequencing depth and normalized via proportions prior to these analyses[73].

## Shotgun metagenomic sequencing and analysis of soil microbiome

Based on the results of the above amplicon sequencing, 12 soil samples that 4 from pH 4.0–4.5, 4 from pH 4.5–5.0 and 4 from pH 5.0–6.0, were selected for shotgun metagenomic sequencing to evaluate the microbial community function. To obtain sufficient metagenomic DNA (2 μg per sample) for all replicates, 4–6 extractions per sample were conducted using the FastDNA SPIN Kit for soil (MP Biomedicals, Santa Ana, CA, USA), and were pooled. Metagenomic libraries were then constructed using a TruSeq™ DNA PCR-free Sample Prep Kit (Illumina, USA) according to the manufacturer's instructions. The metagenomic libraries were sequenced on a HiSeq 2500 sequencer (Illumina, USA), and 150-bp paired-end reads were generated. Approximately 182 GB raw reads were generated after Illumina sequencing, with a total of 899,297,440 resulting paired sequence reads for all 12 soil samples (Supplementary Table 4).

The generated sequence reads were inspected for quality control through the following software programs: SeqPrep (https://github.com/jstjohn/SeqPrep) for tripping non-biological bases in reads, such as primers or barcodes, and Sickle (https://github.com/najoshi/sickle) for filtration of reads whose length after tripping was no more than 50 bp, and whose average quality score was no more than 20. Then, 72.0–82.2 million reads per sample were obtained (Supplementary Table 4). The optimized sequence reads were assembled de novo by SOAPdenovo (http://soap.genomics.org.cn/, Version 1.06) based on a de Bruijn graph for obtaining contigs, and a total of 1,280,885 contigs were generated (Supplementary Table 5). Resulting contigs >200 bp in length were selected to predict open reading frames (ORFs) using MetaGene (http://metagene.cb.k.u-tokyo.ac.jp/), and those for which more than 90% of their length could be aligned to another gene with more than 95% identity (no gaps allowed) were removed as redundancies excepted for the longest gene, resulting in a non-redundant gene catalog comprised of 1,048,576 genes with an average length of 653.16 bp. The high-quality reads from each sample were aligned against the gene catalog by SOAPaligner (http://soap.genomics.org.cn/) with the criterion "identity >95%". We aligned putative amino acid sequences, which were translated from the gene catalog, against the proteins/domains in KEGG databases (Release 79.0) by using BLASTP (BLAST + 2.12.0, http://blast.ncbi.nlm.nih.gov/Blast.cgi) (e value ≤ 1e−5). A total of 1,323,289 genes were hit in the KEGG databases and were assigned to 7234 KEGG Orthology (KO) functional categories and 453 KEGG pathways (Supplementary Fig. 3). We aligned putative amino acid sequences translated from the gene catalog against the proteins/domains in the NCBI-NR database using BLASTP (BLAST + 2.12.0). Genes were taxonomically annotated using corresponding taxonomic information from the NR database. In each sample, the mapped reads of each taxon were counted as the number of taxon-mapped reads.

## Metabolomic analysis of antifungal substances by soil microbiome

Since our results indicated that microbial volatile compounds emitted from soils were involved in fungal pathogen suppression, we further determined the identity of volatile compounds with potential suppression by the bacterial communities using glass Petri-dishes with 'chimney' lids, in which steel traps with 150 mg Tenax TA and 150 mg Carbopack B (Markes International, Llantrisant, UK) were fixed[74]. To test the antifungal effect of microbial volatiles from soils of different pH, collection of antifungal volatiles was done for microbial communities from four independent soil samples of pH 4.0–4.5, 4 from pH 4.5–5.0 and 4 from pH 5.0–6.0, respectively. Sterile Petri-dishes with autoclaved soil served as controls. Volatiles were collected for 24 h after 7 d of incubation.

Next, the traps were removed, capped, and stored at 4 °C until analysis using GC-Q-TOF QTOF (model Agilent 7890B GC and the Agilent 7200 A QTOF, Santa Clara, CA, USA). Detailed information on volatile analysis is given by Hol et al.[67]. For an overall representation of volatile profiles of the microbial communities a partial least square-discriminant analysis (PLS-DA) was made based on peak areas with 95% confidence regions[75]. Identification of metabolites was performed using NIST-MS Search and accurate mass, retention indices, and spectra match factor using NIST (National Institute of Standards and Technology, USA, http://www.nist.gov). The linear retention indexes (lri) values were compared with those found in the NIST database. Some identified compounds were also verified by comparing mass spectra and lri values of pure compounds.

## Statistical analysis

Optimized model for disease severity with physicochemical variables was performed by stepwise regression using forward selection criteria ($p$ of 0.05 for entering and 0.1 for removal). Nonmetric multidimensional scaling (NMDS) of Bray-Curtis dissimilarities was performed using the "vegan" package and "*metaMDS*" function in R software (v3.6.1). Community Shannon diversity was determined by the Shannon-Weaver index, calculated using "vegan" package. Two-sided Wilcoxon rank-sum test was performed between two groups if the distribution of dependent variables did not meet the requirement of normality.

Clustering analysis and heatmap analysis were performed with relative abundance data square-rooted (*sqrt*) transformed using "pheatmap" package. The ordinary least-squares (OLS) regression model was also performed to assess linear the relationship between root rot severity and the taxonomic groups, and the "ggplot2" package was used for data visualization. Further, random forest (RF) modelling was conducted to quantitatively identify the key drivers of DSI among bacterial and fungal phyla using the "randomForest" and "rfPermute" packages.

Based on the results of RF analysis, the significant predictors were further chosen to perform network analysis. The co-occurrence patterns of the bacterial communities and disease severity were constructed based on the Spearman correlation matrix to explore the interactive patterns between microbial taxa in complex communities and their effects on disease occurrence. Besides, those bacterial genera (sum of the relative abundance no less than 1% among all samples) that occurred in more than half of the sample size were constrained genera filtration to reduce rare genera in the dataset. Then, the "Hmisc" package in R was used to analyze the preprocessed data. Only the results with a cut-off at an absolute $r$ value above 0.3 and a $P$ value below 0.05, after adjustment by Benjamini–Hochberg's method to reduce the chances of obtaining false-positive results, were considered statistically robust between taxa, and retained for further network

visualization. The modules were defined as clusters of closely interconnected nodes (e.g., groups of co-occurring or coevolving microbes)[76]. After that, the topological properties of the co-occurrence network, e.g., the numbers of positive and negative correlations, average path length, graph density, network diameter, average clustering coefficient, average degrees, and modularity, were calculated via the "igraph" package in R. For network modules, the module eigengene of a module could summarize the closely connected members within a module[77]. Later on, those genera that showed statistically significant correlations with disease severity were kept for further network visualization. Finally, for the overall network, the top ten taxa among this network with the highest degree, highest closeness centrality, and lowest betweenness centrality were selected as the keystone taxa[78].

### Reporting summary

Further information on research design is available in the Nature Portfolio Reporting Summary linked to this article.

## Data availability

The raw reads from Illumina sequencing and Shotgun metagenomic sequencing described in this study, are available at NCBI under the accession no. PRJNA852869 and PRJNA942228, respectively. Source data are available in the Figshare database (https://doi.org/10.6084/m9.figshare.23813400)[79]. Source data are provided with this paper.

## Code availability

Codes for statistical analyses are available in the Figshare database (https://doi.org/10.6084/m9.figshare.23803443)[80].

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

## Acknowledgements

We thank Prof. Wietse de Boer of NIOO-KNAW for the comments on the manuscript, anonymous farmers for providing permission to conduct fieldwork and Dr. Jia Liu from Jiangxi Academy of Agricultural Science for facilitating the fieldwork on soil survey. This study would not have been possible without the contribution of exceptional fieldworkers. We thank the anonymous reviewer(s) for their contribution to the peer review of this work. This research was supported by the National Natural Science Foundation of China 32122056, 42011045 (X.L.), the National

Key Research and Development Program of China 2017YFD0200604 (Y.D.), Jiangsu Special Fund on Technology Innovation of Carbon Dioxide Peaking and Carbon Neutrality BE2022420 (X.L.), the China Agriculture Research System of MOF and MARA CARS-13 (X.W.), and the Science and Technology Project of Hubei Province of China Tobacco Corporation 027Y2022-001 (Y.D.). M.D-B. acknowledges the contribution from TED2021-130908B-C41/AEI/10.13039/501100011033/Unión Europea NextGenerationEU/PRTR Spanish Ministry of Science and Innovation and PID2020-115813RA-I00/AEI/10.13039/501100011033 of the Spanish Ministry of Science.

## Author contributions

X.L. and D.C. conceived the project, designed the experiments, and conducted the field work. D.C. conducted the laboratory experiments. M.D-B. implemented methodology. D.C. and X.L. analyzed the data, and wrote an initial draft of the manuscript with X.W., Y.D., T.Z., S.Y., V.J.C., D.R., and M.D-B. All authors have discussed the results, read and approved the contents of the manuscript.

## Competing interests

The authors declare no competing interests.
