## [Peer Review File · Nature Communications]

Reviewers' Comments:

Reviewer #1:

Remarks to the Author:

The work "Acidification suppresses the natural capacity of soil microbiome to fight root pathogen infections", presented by Li and coauthors, describes how low soil pH correlates with changes in bacterial and fungal communities and with increase of soil borne infections in field conditions. The authors surveyed 60 peanut fields in China, combining taxonomical with functional approaches to show how pH is correlated with disease incidence. They employed a set of in vitro and in planta experiments allowed the identification of bacterial taxa, metabolites and functions associated with disease suppression. In general, the authors used a very robust approach to address the problem. In fact, it is well known that pH is one of the main drivers for soil microbiome assembly, but I agree with the authors that the knowledge on how this affects soil born diseases and plant health is still limited. Therefore, I recognize the importance of the study. Below are some specific comments:

pH | Recently Luan et al. PNAS 2023 (<https://doi.org/10.1073/pnas.2207832120>) integrated pH into the metabolic theory of ecology to predict bacterial diversity in soil. This paper could be used to enrich the discussion on how soil acidification correlates with bacterial richness (Figs 2 and S8).

Sampling | Clarify whether soil cores were collected in the inter-row or in the planting line (close to rhizosphere).

Figure 2 | What is the legend/meaning for different shapes on panels c and d? Add x-axis title in 'e' and 'f' (as in 'g' and 'h'). The number of observations/samples (n) should be mentioned in the legend. In panels 'e' to 'h', mention the number of samples for each pH range? What is the meaning of ellipses in 'c'? And why they are absent in 'd'? All Figures legends (also in the Supplementary Material) should be reviewed to clarify, the number of observations, the statistical test used, the levels of significance considered etc.

Line 867 | Replace '... "ns" means insignificant difference' by '... "ns" means not significant difference'.

Soil degradation | While the use of "soil degradation" is understandable given the low pH and occurrence of disease, the concept of soil degradation is broader. The peanut yield data would be important to make this point. I would suggest to rephrase the subtitles in the Lines 94 and 113.

Figure 3 | What is meant by 'lineages sensitive to soil acidification'? (See Lines 161 and 166) Are they significantly less or more abundant in low pH? This should be made clear across the text and in the figure legend.

Lack of reps for sequencing | The authors decided to have a single mixed sample for sequencing per field. However, the homogeneity of soil samples in the same field doesn't justify the lack of replications for sequencing.

Line 539 | What were the criteria used to select the 12 fields?

Line 585 | Is 'respectively' for 16S and ITS? Clarify in the text.

Amplicon analysis | Considering that DADA2 pipeline has a better resolution and accuracy than previous pipelines, why did authors use OTUs instead of ASVs? How was the data normalized?

Lines 590-595 | Most of the info in this paragraph concerns the results and not the methodology.

Supplementary Figures | In general, all figures' legends are poorly described. In some figures, statistical tests are even missing, making comparisons impossible.

Reviewer #2:

Remarks to the Author:

The topic that soil-borne pathogens are increasing in agricultural soils is important, and since decades under intense research. Many factors were identified that contributes to the complex problem: missing crop rotation, susceptible cultivars, high pesticide and fertilizer applications, missing (microbial) biodiversity etc. The indigenous seed microbiome plays a role as well because it contributed approx half of the indigenous plants microbiota. Unfortunately, all this is not considered in the manuscript.

The authors focus here on one factor – this is soil acidification, which is a well-known problem of degraded soils. The experimental set up and presentation of results is appropriate and shows that soil acidification is correlated with plant health. This is not really new (pH was found as driver of the soil microbiome in many studies) but evidenced here. However, that fungi are favored under low pH is text book knowledge,

Another problem of the manuscript is that there are huge differences between the rhizosphere and soil. This is mixed in the manuscript but plays a crucial role in pathogen defense, and should be differentiated here.

References on the topic are missing; a lot of opinion, papers and those of the former group of PI are cited but substantial references are missing. This is especially important for the following facts: soil/rhizosphere, mode of action of plant beneficial bacteria, plant microbiome.

Here are only a few examples:

Thomashow LS, Kwak YS, Weller DM. Root-associated microbes in sustainable agriculture: models, metabolites and mechanisms. *Pest Manag Sci*. 2019 Sep;75(9):2360-2367. doi: 10.1002/ps.5406.

Thomashow LS, LeTourneau MK, Kwak YS, Weller DM. The soil-borne legacy in the age of the holobiont. *Microb Biotechnol*. 2019 Jan;12(1):51-54. doi: 10.1111/1751-7915.13325.

Schlatter D, Kinkel L, Thomashow L, Weller D, Paulitz T. Disease Suppressive Soils: New Insights from the Soil Microbiome. *Phytopathology*. 2017 Nov;107(11):1284-1297. doi: 10.1094/PHYTO-03-17-0111-RVW.

Wolfgang A, Zachow C, Müller H, Grand A, Temme N, Tilcher R, Berg G. Understanding the Impact of Cultivar, Seed Origin, and Substrate on Bacterial Diversity of the Sugar Beet Rhizosphere and Suppression of Soil-Borne Pathogens. *Front Plant Sci*. 2020 Sep 30;11:560869. doi: 10.3389/fpls.2020.560869.

Xu S, Liu YX, Cernava T, Wang H, Zhou Y, Xia T, Cao S, Berg G, Shen XX, Wen Z, Li C, Qu B, Ruan H, Chai Y, Zhou X, Ma Z, Shi Y, Yu Y, Bai Y, Chen Y. *Fusarium* fruiting body microbiome member *Pantoea agglomerans* inhibits fungal pathogenesis by targeting lipid rafts. *Nat Microbiol*. 2022 Jun;7(6):831-843. doi: 10.1038/s41564-022-01131-x.

Chapelle E, Mendes R, Bakker PA, Raaijmakers JM. Fungal invasion of the rhizosphere microbiome. *ISME J*. 2016 Jan;10(1):265-8. doi: 10.1038/ismej.2015.82.

Soil-borne pathogens: there are much more than *Fusarium* but only one *Fusarium* species was studied. The pathogenicity of strains depends not only on pH, also on overall microbial diversity of the systems. Under specific conditions, and in nature, a lot of *Fusarium* strains show plant beneficial functions! In this sense, the title should be specified.

Altogether, although I like the presentation of results, this manuscript should be substantially and critically revised, over-interpretation avoided and relevant literature cited.

Responses to reviewers' comments

Reviewer #1 (Remarks to the Author):

The work “Acidification suppresses the natural capacity of soil microbiome to fight root pathogen infections”, presented by Li and coauthors, describes how low soil pH correlates with changes in bacterial and fungal communities and with increase of soil borne infections in field conditions. The authors surveyed 60 peanut fields in China, combining taxonomical with functional approaches to show how pH is correlated with disease incidence. They employed a set of in vitro and in planta experiments allowed the identification of bacterial taxa, metabolites and functions associated with disease suppression. In general, the authors used a very robust approach to address the problem. In fact, it is well known that pH is one of the main drivers for soil microbiome assembly, but I agree with the authors that the knowledge on how this affects soil borne diseases and plant health is still limited. Therefore, I recognize the importance of the study. Below are some specific comments:

Reply: Thank you for positive and constructive comments, which have helped us to improve our research. We aimed to address all your comments below.

pH | Recently Luan et al. PNAS 2023 (<https://doi.org/10.1073/pnas.2207832120>) integrated pH into the metabolic theory of ecology to predict bacterial diversity in soil. This paper could be used to enrich the discussion on how soil acidification correlates with bacterial richness (Figs 2 and S8).

Reply: We thank reviewer for the reference. We now cited this paper, and have included this important reference in our discussion section:

“Recently, Luan et al.³⁶ mechanistically integrated pH into the metabolic theory of ecology to predict soil bacterial diversity, and found that the integrated model reliably and accurately predicted the patterns of bacterial diversity across soil pH gradients. Acidic conditions can disrupt the energy metabolism of soil bacterial taxa by reducing their proton gradients, which can impede the survival of bacterial taxa that have high energy demands or low tolerance to acidic conditions”. (Lines 316-322)

Sampling | Clarify whether soil cores were collected in the inter-row or in the planting line (close to rhizosphere).

Reply: We have now clarified that samples were collected in the inter-row. Clarified in lines 433.

Figure 2 | What is the legend/meaning for different shapes on panels c and d? Add x-axis title in ‘e’ and ‘f’ (as in ‘g’ and ‘h’). The number of observations/samples (n) should be mentioned in the legend. In panels ‘e’ to ‘h’, mention the number of samples for each pH range? What is the meaning of ellipses in ‘c’? And why they are absent in ‘d’? All Figures legends (also in the Supplementary Material) should be reviewed to clarify,

the number of observations, the statistical test used, the levels of significance considered etc.

Reply: Thanks for your suggestion for polishing our figures and figure legend. We have revised them as follows:

--Different shapes denote different pH categories. We have added notice in Fig. 2d. Moreover, we have supplemented text in the figure legend as “Ranges of pH are represented by different shapes in panels (c) and (d).” (Lines 953-954)

--We have added x-axis title as 4.0-4.5, 4.5-5.0, 5.0-6.0, and 6.0-7.0 of pH range in Fig. 2e, f.

--We have added the total number of observations as “n = 180” in the figure legend (Lines 946, 952), and “n = 60” in lines 953, 953, 957.

--The numbers of observations for each pH range have been added in Fig. 2e, f, g, h, as well as Figure S8.

-- Due to the stress value of bacterial community being less than 0.08 (Fig. 2c), implying a more reliable interpretation of the results than the fungal community plot with stress value more than 0.09 (Fig. 2d), we originally intended to add an ellipse to characterize the obvious clustering of samples in the four pH ranges. For the sake of objectivity, as well as consistency with plot d (Fig. 2d), we have removed the ellipse.

-- Significant correlation coefficient is noted by asterisks based on Spearman’s method. P values were adjusted by Benjamini–Hochberg false discovery correction (*P < 0.05, **P < 0.01, ***P < 0.001; n = 180). (Lines 944-946)

--The statistical test used is F-test, and P < 0.001 denotes the significance of the model and the significance of the predictor in the model (n = 180). (Lines 950-952)

-- Boxplots indicate median (middle line), 25th, 75th percentiles (box), and 5th and 95th percentiles (whiskers). (Lines 957-958)

-- Numbers in brackets denote the sample size in corresponding boxplot. (Lines 960-961)

-- Ordinary least squares (OLS) linear regression between (Lines 974-975)

--The statistical test used is F-test, and P < 0.05 denotes the overall significance of the regression model. (Lines 979-980)

-- The blue fitted lines are from OLS linear regression between the suppression rate and soil pH (n = 180), and the shaded areas indicate 95% confidence interval of the fit. The statistical test used is F-test, and P < 0.001 denotes the overall significance of the regression model (n= 180). (Lines 992-995)

-- Boxplots indicate median (middle line), 25th, 75th percentiles (box), and 5th and 95th percentiles (whiskers). Asterisks indicate significant differences as represented by the Wilcoxon test (*P < 0.05, **P < 0.01, ***P < 0.001), and “ns” means not significant difference. Numbers in brackets denote the sample size in corresponding boxplot (n = 45 for each pH category). (Lines 998-1003)

Supplementary Materials:

--Figure S1 (n = 60 study fields). Yichun, Nanchang, Fuzhou, Shangrao, and Yingtan are cities in southeast China. Dots denote the field locations of paired plant and soil sampling. (Lines 52-54)

--Figure S8 Effects of soil acidification on Chao1 richness of the bacterial (a) and fungal (b) communities from soil samples of different pH ranges (n = 60). Boxplots indicate median (middle line), 25th, 75th percentiles (box), and 5th and 95th percentiles (whiskers). (Lines 95-97)

--Figure S8 Numbers in brackets denote the sample size in corresponding boxplot. (Line 98)

--Figure S10 Ordinary least squares linear regression between the disease severity index and microbial diversity (n = 60). The statistical test used is F-test, and $P < 0.05$ denotes the overall significance of the regression model. The shaded areas indicate the 95% confidence interval of the regression fit. (Lines 107-109)

--Figure S12 (n=60) (Line 129)

Line 867 | Replace ‘... “ns” means insignificant difference’ by ‘...“ns” means not significant difference’.

Reply: Revised as “ns” means not significant difference. (Line 960) (Supplementary Information Lines 97-98)

Soil degradation | While the use of "soil degradation" is understandable given the low pH and occurrence of disease, the concept of soil degradation is broader. The peanut yield data would be important to make this point. I would suggest to rephrase the subtitles in the Lines 94 and 113.

Reply: Thanks for your suggestion. We have changed “soil degradation” to “soil acidification”. (Lines 98, 117)

Figure 3 | What is meant by ‘lineages sensitive to soil acidification’? (See Lines 161 and 166) Are they significantly less or more abundant in low pH? This should be made clear across the text and in the figure legend.

Reply: Yes. We have revised it as follows:

“a phylogenetic tree was constructed to depict dominant (present in over 25% fields) microbial taxa that exhibit significant responses to soil acidification. The tree indicated a nonrandom phylogenetic distribution of microbial taxa in response (Fig. 3c)”. (Lines 173-175)

Figure legend of Fig. 3c was revised to: *“Phylogenetic tree depicting microbial taxa exhibiting significant responses to soil acidification”.* (Lines 965-966)

Lack of reps for sequencing | The authors decided to have a single mixed sample for sequencing per field. However, the homogeneity of soil samples in the same field doesn't justify the lack of replications for sequencing.

Reply: We are also very sorry for the inaccurate and unclear in the Methods section. We have now clarified that:

1. In the case of our field survey, three quadrats (replicates) in each field were arranged randomly. Within each quadrat, we collected plant and soil paired samples. Our sampling resulted in 180 soil samples (10 soil cores per sample) and 180 plant samples (50 individuals per sample) from 60 peanut fields (60 fields x 3 quadrates).

Thus, our statistical analyses are based on soil and plant variables across these 180 quadrats. These have been specified in the Materials and Methods part (Lines 426-431).

2. For the subsequent soil microbiome sequencing, the sampled soils were classified into four categories (*i.e.*, 4.0-4.5, 4.5-5.0, 5.0-6.0, and 6.0-7.0) based on soil pH, and soil samples falling into the corresponding pH category were adopted as independent replicates (see details for figures 2-3, and supplementary figures S8, S9, and S13). Consequently, genomic DNA extracted from three soil samples per field was pooled, resulting in 60 composite DNA samples corresponding to the 60 crop fields for subsequent amplicon sequencing. For more clarity of our study schedule, we have specified the methodology in details in the Materials and Methods part (Lines 476-486).

3. For Illumina sequencing, DNA PCR amplification was triplicated for each sample using the ABI GeneAmp® 9700 Thermal Cycler (Thermo Fisher Scientific, USA) following standardized protocols. Subsequently, the PCR products were pooled for sequencing analysis on an Illumina MiSeq PE 250 sequencer. For more clarity of our amplification schedule, we have added the methodology in details in the Materials and Methods part (Lines 502-512).

Line 539 | What were the criteria used to select the 12 fields?

Reply: We selected 4 field soil samples within the pH range of 4.0-4.5, 4.5-5.0, and 5.0-6.0, respectively, with the following selection criteria: 1) different pH values; 2) yet similar other physicochemical properties in each pH category. This has been detailed in Lines 606-607.

Line 585 | Is 'respectively' for 16S and ITS? Clarify in the text.

Reply: Yes. We have specified it as "for 16S and ITS, respectively". (Line 118)

Amplicon analysis | Considering that DADA2 pipeline has a better resolution and accuracy than previous pipelines, why did authors use OTUs instead of ASVs? How was the data normalized?

Reply: We appreciate your comment and suggestion regarding the use of OTUs in our study.

1. First, we would like to highlight that both OTUs and ASVs are known to provide very similar patterns in microbial ecology. See Glassman SI, Martiny JBH. Broad-scale Ecological Patterns Are Robust to Use of Exact Sequence Variants versus Operational Taxonomic Units. *mSphere*. 2018, 3:e00148-18. doi: 10.1128/mSphere.00148-18. for detailed evaluation.

2. In fact, as this reviewer might know, amplicon sequencing is limited to a small fraction of the complete gene (e.g., here over 400bp out of ~1600bp for 16S sequencing). In this respect, using 100% ASV or 97% OTUs does not really change much when it comes to the resolution of the method, as supported by the paper from Glassman & Martiny above. This is an important point normally neglected in the ASV literature.

3. Also, in our study, we focused more on identifying broad taxonomic groups of soil communities across the pH gradients, and did not involve “species” identification. Likewise, similar studies conducted by Yuan et al. (2021), and Zhang et al. (2022) adopted OTUs in their microbiome sequence inference as well. Accordingly, we have revised it as: “Given that we aimed to identify broad taxonomic groups of soil microbial communities across samples of a pH gradient, the sequences retained for each sample were then clustered and assigned to OTUs at ...”. (Lines 649-652)

4. Finally, we would like to clarify that we used quality trimming to remove low-quality bases and ambiguous bases, and used the rarefaction method to normalize the sequence data to an even depth by randomly subsampling the lowest number of reads across all samples. This allows for a fair comparison of microbial diversity between samples with different sequencing depths. Some details have been added in the Method section (Lines 643-646, 656-658).

References:

- Glassman, S.I., Martiny, J.B.H. Broadscale ecological patterns are robust to use of exact sequence variants versus operational taxonomic units. *mSphere* **3**, e00148-18 (2018).
- Schloss, P. D. Amplicon sequence variants artificially split bacterial genomes into separate clusters. *mSphere* **6**, e00191-00121 (2021).
- Yuan, M. M. et al. Climate warming enhances microbial network complexity and stability. *Nat. Clim. Change* **11**, 343–348 (2021).
- Zhang, L. et al. A highly conserved core bacterial microbiota with nitrogen-fixation capacity inhabits the xylem sap in maize plants. *Nat. Commun.* **13**, 3361 (2022).

Lines 590-595 | Most of the info in this paragraph concerns the results and not the methodology.

Reply: We have moved this paragraph to the Results section. (Lines 118-126)

Supplementary Figures | In general, all figures’ legends are poorly described. In some figures, statistical tests are even missing, making comparisons impossible.

Reply: Thanks for your suggestion. We have carefully revised the figure legends in Supplementary Figures for clearer description, including statistical tests and number of observations.

Reviewer #2 (Remarks to the Author):

The topic that soil-borne pathogens are increasing in agricultural soils is important, and since decades under intense research. Many factors were identified that contributes to the complex problem: missing crop rotation, susceptible cultivars, high pesticide and fertilizer applications, missing (microbial) biodiversity etc. The indigenous seed microbiome plays a role as well because it contributed approx half of the indigenous plant microbiota. Unfortunately, all this is not considered in the manuscript.

Reply: Thanks! We have updated our discussion to account for this important point:

“Finally, we posit that it is fundamental to consider the broader context of crop disease management. Several factors, such as the absence of crop rotation practices, the use of susceptible cultivars, the excessive application of pesticides and fertilizers, and the abandonment of indigenous seed microbiomes have been identified as contributors to the greater occurrence of soil-borne diseases in agricultural ecosystems^{57, 58}. Our results revealed that the native soil microbiome plays an essential role in regulating the defense capacity of plants against fungal pathogens. Changes in the soil microbiome associated with acidification linked with agricultural management can have important consequences for plant disease severity^{59, 60}, and we provided experimental and mechanistic explanations on why this is the case. Our results further highlight the role of soil-borne microbial legacies in shaping plant health for food production⁹. The complex interactions between plants and soil microbiome are still being tackled, with important work now highlighting microbial-driven defense mechanisms against pathogen infections⁶¹. Further investigations considering the influence of the soil microbiome and its legacy on pathogen infection are needed to better understand the future of food production under global environmental change.”
(Lines 383-398)

The authors focus here on one factor – this is soil acidification, which is a well-known problem of degraded soils. The experimental set up and presentation of results is appropriate and shows that soil acidification is correlated with plant health. This is not really new (pH was found as driver of the soil microbiome in many studies) but evidenced here. However, that fungi are favored under low pH is text book knowledge.

Reply: Thanks for your comments. Both bacteria and fungi have been acknowledged with sensitive responses to pH of soil environment. We believe that soil fungi are often favored under low pH conditions mainly due to their metabolic characteristics and adaptations to acidic environments. This is especially true in natural ecosystem wherein fungal hyphae are intact. However, our study is based on disturbed arable soils wherein bacterial communities are known to dominate these soils (Heijden et al. 2008). Moreover, in our study we did not find any correlation between fungal communities and pH, challenging previous expectations. Our multiple field surveys, experiments and assays provide robust evidence that the capacity of bacterial communities to regulate plant disease is pH dependent. Acidic conditions can disrupt the energy metabolism of soil bacterial taxa by reducing their proton gradients, which can impede the survival of bacterial taxa with have high energy demands or low tolerance to acidic conditions (Rousk et al. 2010; Park and Park 2018; Luan et al. 2023). This new information is novel and helpful to understand plant disease in a context of management as highlighted by reviewer #1.

Reference:

- Heijden, M.G., et al. The unseen majority: soil microbes as drivers of plant diversity and productivity in terrestrial ecosystems. *Ecol Lett.* **11**(3), 296-310 (2008).
- Rousk, J. et al. Soil bacterial and fungal communities across a pH gradient in an arable soil. *ISME J.* **4**, 1340-1351 (2010).

Park, C. & Park, W. Survival and energy producing strategies of alkane degraders under extreme conditions and their biotechnological potential. *Front. Microbiol.* **9**, 1081 (2018).
Luan, L. et al. Integrating pH into the metabolic theory of ecology to predict bacterial diversity in soil. *Proc. Natl Acad. Sci. USA* **120**, e2207832120 (2023).

Another problem of the manuscript is that there are huge differences between the rhizosphere and soil. This is mixed in the manuscript but plays a crucial role in pathogen defense, and should be differentiated here.

Reply: We appreciate the reviewer's comment regarding the differentiation between the rhizosphere and bulk soil. Our study focused on the overall impact of soil acidification on soil microbiomes and the associated soil suppression on pathogen development. For clarity, we have carefully checked our expressions throughout the manuscript to avoid the mixed notions in the roles of bulk soil that are different from plant rhizosphere.

References on the topic are missing; a lot of opinion, papers and those of the former group of PI are cited but substantial references are missing. This is especially important for the following facts: soil/rhizosphere, mode of action of plant beneficial bacteria, plant microbiome.

Here are only a few examples:

Thomashow LS, Kwak YS, Weller DM. Root-associated microbes in sustainable agriculture: models, metabolites and mechanisms. *Pest Manag Sci.* 2019 Sep;75(9):2360-2367. doi: 10.1002/ps.5406.

Thomashow LS, LeTourneau MK, Kwak YS, Weller DM. The soil-borne legacy in the age of the holobiont. *Microb Biotechnol.* 2019 Jan;12(1):51-54. doi: 10.1111/1751-7915.13325.

Schlatter D, Kinkel L, Thomashow L, Weller D, Paulitz T. Disease Suppressive Soils: New Insights from the Soil Microbiome. *Phytopathology.* 2017 Nov;107(11):1284-1297. doi: 10.1094/PHYTO-03-17-0111-RVW.

Wolfgang A, Zachow C, Müller H, Grand A, Temme N, Tilcher R, Berg G. Understanding the Impact of Cultivar, Seed Origin, and Substrate on Bacterial Diversity of the Sugar Beet Rhizosphere and Suppression of Soil-Borne Pathogens. *Front Plant Sci.* 2020 Sep 30;11:560869. doi: 10.3389/fpls.2020.560869.

Xu S, Liu YX, Cernava T, Wang H, Zhou Y, Xia T, Cao S, Berg G, Shen XX, Wen Z, Li C, Qu B, Ruan H, Chai Y, Zhou X, Ma Z, Shi Y, Yu Y, Bai Y, Chen Y. *Fusarium* fruiting body microbiome member *Pantoea agglomerans* inhibits fungal pathogenesis by targeting lipid rafts. *Nat Microbiol.* 2022 Jun;7(6):831-843. doi: 10.1038/s41564-022-01131-x.

Chapelle E, Mendes R, Bakker PA, Raaijmakers JM. Fungal invasion of the rhizosphere microbiome. *ISME J.* 2016 Jan;10(1):265-8. doi: 10.1038/ismej.2015.82.

Reply: We thank reviewer for the suggestion and the detailed references. We have combined the comment with the above references to form a discussion for addressing the related topic. (Lines 383-398)

Soil-borne pathogens: there are much more than *Fusarium* but only one *Fusarium*

species was studied. The pathogenicity of strains depends not only on pH, also on overall microbial diversity of the systems. Under specific conditions, and in nature, a lot of *Fusarium* strains show plant beneficial functions! In this sense, the title should be specified.

Reply: Thank you for your comments. We have specified the title as “Acidification suppresses the natural capacity of soil microbiome to fight pathogenic *Fusarium* infections.” (Lines 1-2), and some descriptions for *Fusarium* in the abstract as well. (Line 34)

Altogether, although I like the presentation of results, this manuscript should be substantially and critically revised, over-interpretation avoided and relevant literature cited.

Reply: Thanks for your appreciation and suggestions. We have critically revised our manuscript, specified our expression (e.g., sample in the inter-row, bulk soil, indigenous microbiome, title, etc.), cited the relevant literature, and added some discussion as well. Please see details in the Revised Manuscript with changes marked. Thank you again for the opportunity of polishing our manuscript.

Reviewers' Comments:

Reviewer #1:
None

Reviewer #2:
None